# Posted Price Mechanisms for Online Allocation with Diseconomies of Scale

## Abstract

This paper addresses the online $k$-selection problem with diseconomies of scale (OSDoS), where a seller seeks to maximize social welfare by optimally pricing items for sequentially arriving buyers, accounting for increasing marginal production costs. Previous studies have investigated deterministic dynamic pricing mechanisms for such settings. However, significant challenges remain, particularly in achieving optimality with small or finite inventories and developing effective randomized posted price mechanisms. To bridge this gap, we propose a novel randomized dynamic pricing mechanism for OSDoS, providing a tighter lower bound on the competitive ratio compared to prior work. Our approach ensures optimal performance in small inventory settings (i.e., when $k$ is small) and surpasses existing online mechanisms in large inventory settings (i.e., when $k$ is large), leading to the best-known posted price mechanism for optimizing online selection and allocation with diseconomies of scale across varying inventory sizes.

**ACM Reference Format:**
Anonymous Author(s). 2024. Posted Price Mechanisms for Online Allocation with Diseconomies of Scale. In . ACM, New York, NY, USA, 19 pages. https://doi.org/10.1145/nnnnnnn.nnnnnnn

## 1 Introduction

Online resource allocation has been widely studied in recent years and finds a broad range of applications in cloud computing [24, 25], network routing [2, 7, 8], and various other online, market-based Internet platforms. In this problem, most existing studies assume that the seller has a finite inventory of resources before a stream of online buyers arrives, with the goal of maximizing social welfare or profit from these resources. However, in real-world applications, sellers often face diseconomies of scale in providing resources—meaning they incur increasing marginal costs for supplying each additional unit of resource. For instance, in cloud computing systems, the power cost of servers increases superlinearly as the utilization of computing resources grows [1]. Similarly, in network routing, congestion costs (e.g., end-to-end delay) increase significantly with the rise in traffic intensity brought by users.

In this work, we study online resource allocation with increasing marginal production costs. In particular, we frame it as an online $k$-selection with diseconomies of scale (OSDoS) in a posted price mechanism: A seller offers a certain item to buyers arriving one at a time in an online manner. Each buyer has a private valuation $v_t$ for

one unit of the item. The seller can produce $k$ units of the item in total; however, the marginal cost of producing each unit increases as more units are produced. When the $t$-th buyer arrives, the seller posts a price $p_t$ to the buyer, provided that fewer than $k$ units have already been produced and allocated. If the buyer's valuation $v_t$ exceeds $p_t$, the buyer accepts the price and takes one unit of the item. The objective is to maximize social welfare, defined as the sum of the utilities of all the buyers and the revenue of the seller.

The incorporation of increasing marginal production costs in online resource allocation was initially pioneered by [5] and later studied by [12] in the context of online combinatorial auctions. Since then, different variants of OSDoS have been studied, such as online convex packing and covering [3], online knapsack with packing costs [22], and online selection with convex costs [23]. The common crux of these problems is that if, at early stages, the pricing of the resource is too aggressive (i.e., the price is set too low), a large number of items may be allocated to low-value buyers, leading to an increase in production costs and a lower social welfare. Conversely, if the pricing is too conservative (i.e., the price is set too high) at early stages, the producer may lose the opportunity to sell enough items to buyers. Thus, we must carefully design the prices of these $k$ units to avoid selecting buyers that appear advantageous early on but cause the rapid growth of marginal production costs as more units are produced.

To address above challenge, Huang et al. [12] developed optimal *deterministic dynamic pricing* mechanisms for fractional online combinatorial auctions with production costs and infinite production capacity (i.e., $k = \infty$). They also showed that similar mechanisms could be designed for the integral case based on the fractional pricing functions, achieving a competitive ratio that can be arbitrarily close to that of the fractional setting at the expense of some nonzero additive loss. However, as the competitive ratio in the integral setting approaches the lower bound derived in the fractional setting [12], the additive loss tends to approach infinity, which is undesirable. To overcome this limitation, Tan et al. [23] considered online selection with convex costs and limited supply (i.e., $k$ is finite). Notably, they established a lower bound for the integral setting without incurring any additive loss and further showed that the competitive ratio of the proposed deterministic posted price mechanism asymptotically converges to the lower bound in large inventory settings (i.e., when $k$ approaches infinity). Recently, Sun et al. [20] developed a *randomized static pricing* algorithm that sampled one static price from a pre-determined distribution for solving OSDoS. By utilizing randomization, the static pricing algorithm developed in [20] can outperform the deterministic dynamic pricing algorithm developed in [23] in small inventory settings. However, the algorithm is not asymptotically optimal in large inventory settings, and fails to converge to the lower bound obtained in [23] as $k$ approaches to infinity.

Despite previous efforts, two questions remain unresolved: First, how to derive a tight lower bound for OSDoS in small inventory settings? Second, it remains an open question how to develop randomized algorithms to solve OSDoS with tight guarantees, especially for settings when $k$ is small.

In this paper, we affirmatively answer these two questions by deriving a new tight lower bound for the OSDoS problem, providing the best-known result in both small inventory and asymptotically large inventory settings. Additionally, inspired by the new lower bound results, we develop a novel algorithm that utilizes up to $k$ randomized prices, rather than just one. We demonstrate that our proposed *randomized dynamic pricing* algorithm is optimal for small inventory settings and outperforms existing designs presented in [23] and [20] for large inventory settings.

## 1.1 Overview of Main Results and Techniques

The primary contribution of this paper is the development of novel posted price mechanisms utilizing randomized dynamic pricing schemes that improve upon the results in [5, 12, 20, 23]. The randomized dynamic pricing scheme, termed r-Dynamic, sequentially updates the price of the item as new units are produced and sold. More specifically, as the marginal production cost of the item increases with each additional unit, r-Dynamic employs a different cumulative distribution function (CDF) to generate an independent randomized price for each new unit of the item. The main result concerning the lower bound can be stated as follows:

THEOREM 1 (INFORMAL STATEMENT OF THEOREM 3). *Assume that buyers' valuations are bounded within the range $[L, U]$ and the cumulative cost of production up to the $i$-th unit is given by $f(i)$. The seller can produce a total of $k$ units. For any given $k \geq 1$, $U \geq L \geq 1$, and a cumulative production cost function $f$, no online algorithm can be $(\alpha_S^*(k) - \epsilon)$-competitive for any $\epsilon > 0$, where $S := \{L, U, f\}$.*

We note that [23] also established a lower bound for the competitive ratio of any online algorithm for OSDoS. However, the lower bound in [23] was derived by connecting the integral selection problem to its fractional counterpart. As a result, the cumulative production cost function $f$ in [23] needs to be defined not only at discrete points but also at every fractional point in the range $[0, k]$. This brings two issues: (i) it may be impractical to assume the availability of a continuous cost function $f$ due to the integral nature of the problem, and (ii) the derived lower bound in [23] is only tight under large inventory settings due to the fact that the lower bound was established by assuming $k \to \infty$. In this paper, we address these two issues by deriving the lower bound $\alpha_S^*(k)$ via a totally different approach. In particular, we do not rely on results in the fractional setting and only need to assume that the cost function $f(i)$ is defined at discrete points for $i \in \{1, 2, \cdots, k\}$, leading to the tight lower bound $\alpha_S^*(k)$ for all $k \geq 1$.

THEOREM 2 (INFORMAL STATEMENT OF THEOREM 5). *For any given $k \geq 1$, $U \geq L \geq 1$, and a cumulative production cost function $f$, there exists a randomized dynamic price mechanism (r-Dynamic) that achieves a competitive ratio of $\alpha_S^*(k) \cdot \exp(\frac{\alpha_S^*(k)}{k})$. In particular, when $k = 2$, r-Dynamic is $\alpha_S^*(2)$-competitive.*

Due to the arbitrary nature of the cost function $f$, neither our work nor [23] can derive the competitive ratio in closed form. Thus,

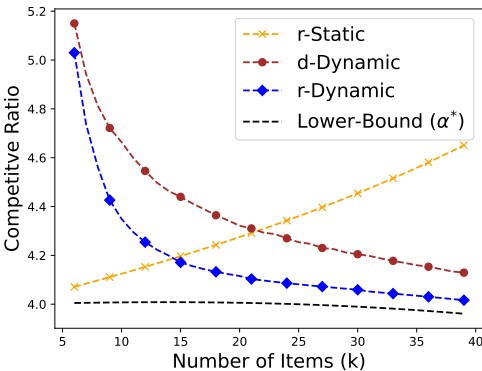

**Figure 1: The blue curve (i.e., r-Dynamic) corresponds to the competitive ratio of Algorithm 1 that uses randomized dynamic pricing. The red curve (i.e., d-Dynamic) and the yellow curve (i.e., r-Static) correspond to the competitive ratios of the deterministic dynamic pricing mechanism developed by [23] and the static randomized pricing mechanism by [20]. In this figure, we set $L = 1$, $U = 10$, and $f(i) = \frac{i^2}{59}$.**

we cannot directly compare the competitive ratio of r-Dynamic with the deterministic dynamic pricing mechanism (d-Dynamic) proposed in [23]. In Figure 1, we compare the asymptotic performance of our r-Dynamic with d-Dynamic in [23] and the randomized static pricing mechanism (r-Static) in [20]. The results show that our proposed r-Dynamic significantly outperforms both d-Dynamic and r-Static, and converges more quickly to the lower bound as $k$ approaches infinity. Furthermore, in small $k$ settings, we prove that r-Dynamic can even attain the lower bound when $k = 2$. In addition to the strong theoretical guarantees, we also demonstrate that the empirical performance of r-Dynamic outperforms both d-Dynamic and r-Static (see Section 4.2), demonstrating that r-Dynamic is a superior algorithm compared to existing designs.

The key technical component in deriving the above lower and upper bound results is a new *representative function*-based approach. This approach represents the dynamics of any randomized online algorithm using a sequence of $k$ probability functions, denoted as $\{\psi_i\}_{\forall i \in [k]}$. Specifically, we design a family of hard instances and then characterize the performance of any $\alpha$-competitive online algorithm on these hard instances using a set of differential equations involving $\{\psi_i\}_{\forall i \in [k]}$. To determine the lower bound of the competitive ratio in Theorem 1, $\alpha_S^*(k)$, we show that it suffices to compute the minimum value of $\alpha$ such that the set of differential equations has a feasible solution, namely, a sequence of valid probability functions $\{\psi_i\}_{\forall i \in [k]}$. By reverse engineering these differential equations, we derive a corresponding set of inverse probability functions, denoted as $\{\phi_i\}_{\forall i \in [k]}$, for pricing each unit of the item, which leads to the r-Dynamic in Theorem 2.

## 1.2 Other Related Work

Online resource allocation, the process of assigning limited resources to a sequence of online requests to maximize social welfare or profit, has long been a central topic in computer science and

operations research. In addition to the related work mentioned earlier, readers are referred to the survey by Gupta and Singla on the secretary problem [11] and the references therein for a detailed discussion of online allocation and selection in random-order models. Significant progress has also been made in recent years on the study of the prophet inequality through the lens of posted price mechanisms [16] [9], as well as on online matching and its applications to Internet advertising (e.g., see [18] and [13]). Beyond the stochastic i.i.d. model in the prophet inequality, a recent line of work explores the correlated arrival model based on a Markov chain [14]. However, all these studies focus on different variants of online allocation and selection problems without accounting for production costs, whereas our primary focus is to examine the impact of increasing marginal production costs on online $k$-selection.

In recent years, there have been attempts to study online allocation problems with different forms of production costs in stochastic settings (e.g., [6], [10], [4], and [19]). Specifically, the authors of [6] considered the problem of online allocation of goods with economies of scale, or decreasing marginal cost per item for the seller. When customers have unit demand and arrive one at a time with general valuations on items sampled i.i.d. from some unknown underlying distribution, the authors of [6] developed a strategy that is constant-competitive with respect to the optimal allocation in hindsight. In comparison, the authors of [19] considered a Bayesian online allocation problem in the presence of convex production costs (i.e., diseconomies of scale). The framework by [19] yields posted price mechanisms with $O(1)$-approximation factors for fractionally subadditive buyers and logarithmic approximations for subadditive buyers. Recall that we consider OSDoS in adversarial settings by assuming no knowledge of the arrival instance (other than the finite support of the valuations), so these results are not directly comparable to ours.

On the applied side, the allocation of limited resources under diseconomies of scale is prevalent across various online platforms in multiple domains. For instance, in online cloud resource allocation with server costs [24], convex and increasing server costs model the energy consumption of each computing server as a function of total CPU utilization. Another example is the online electric vehicle charging problem with electricity costs [21], where electricity generation costs are typically nonlinear and often modeled as a quadratic function of total generation.

## 2 Problem Statement and Assumptions

We formally define online $k$-selection with diseconomies of scale (OSDoS) as follows. Consider an online market operating under posted price mechanisms. On the supply side, a seller can produce a total of $k$ units of an item, with increasing (or at least non-decreasing) marginal production costs. Let $\boldsymbol{c} := \{c_i\}_{\forall i \in [k]}$ represent the *marginal production cost*, where $c_i$ denotes the cost of producing the $i$-th unit, and $c_1 \leq c_2 \leq \cdots \leq c_k$. Define $f(i) = \sum_{j=1}^{i} c_j$ as the *cumulative production cost* of the first $i$ units. On the demand side, $T$ buyers arrive sequentially, each demanding one unit of the item. Let $v_t$ denote the private valuation of the $t$-th buyer. Once buyer $t$ arrives, a price $p_t$ is posted, and then the buyer decides to accept the price and make a purchase if a non-negative utility is gained $v_t - p_t \geq 0$, and reject it otherwise.

Let $x_t \in 0, 1$ represent the decision of buyer $t$, where $x_t = 1$ indicates a purchase and $x_t = 0$ otherwise. Then buyer $t$ obtains a utility $(v_t - p_t)x_t$ and the seller collects a total revenue of $\sum_{t \in [T]} p_t x_t - f(\sum_{t \in [T]} x_t)$ from all buyers. The goal of the online market is to determine the posted prices $\{p_t\}_{\forall t \in [T]}$ to maximize the social welfare, which is the sum of utilities of all the buyers and the revenue of the producer, i.e., $\sum_{t \in [T]} x_t \cdot (v_t - p_t) + \sum_{t \in [T]} x_t \cdot p_t - f(\sum_{t \in [T]} x_t) = \sum_{t \in [T]} v_t x_t - f(\sum_{t \in [T]} x_t)$.

Let $\mathcal{I} = \{v_1, \cdots, v_T\}$ denote an arrival instance of buyers. An optimal offline algorithm that knows all the information of $\mathcal{I}$ can obtain the optimal social welfare $\mathrm{OPT}(\mathcal{I})$ by solving the following optimization problem

$$\mathrm{OPT}(\mathcal{I}) = \max_{x_t \in \{0,1\}} \sum_{t \in [T]} v_t x_t - f\left(\sum_{t \in [T]} x_t\right),$$

$$\text{s.t.} \sum_{t \in [T]} x_t \leq k.$$

However, in the online market, the posted price $p_t$ is determined without knowing the valuations of future buyers $\{v_\tau\}_{\tau > t}$. We aim to design an online mechanism to determine the posted prices such that the social welfare achieved by the online mechanism, denoted by $\mathrm{ALG}(\mathcal{I})$, is competitive compared to $\mathrm{OPT}(\mathcal{I})$. Specifically, an online algorithm is $\alpha$-competitive if for any input instance $\mathcal{I}$, the following inequality holds:

$$\alpha \geq \frac{\mathrm{OPT}(\mathcal{I})}{\mathbb{E}[\mathrm{ALG}(\mathcal{I})]},$$

where the expectation of $\mathbb{E}[\mathrm{ALG}(\mathcal{I})]$ is taken with respect to the randomness of the online algorithm. To attain a bounded competitive ratio, we consider a constrained adversary model [15, 23], where the buyers' valuations are assumed to be bounded.

ASSUMPTION 1. *Buyers' valuations are bounded in $[L, U]$, i.e., $v_t \in [L, U], \forall t \in [T]$.*

The interval $[L, U]$ can be considered as the prediction interval that covers the valuations of all buyers [15], and is known to the online algorithm. As shown in [23], the competitive analysis of online algorithms for OSDoS depends on the relationship between buyers' valuations and the production cost function. For simplicity of presentation, we focus on the case where the production cost is always smaller than the buyer's valuation, i.e., $c_k < L$, and derive lower and upper bound results of OSDoS in Section 3 and Section 4, respectively. We show in Appendix H and I that the assumption of $c_k < L$ is without loss of generality as our lower/upper bound results can be naturally extended to the general case.

## 3 Lower Bound for OSDoS: Hardness of Allocation with Diseconomies of Scale

We begin by deriving a tight lower bound for OSDoS. The insights gained from this analysis guide the design of r-Dynamic (Algorithm 1) in Section 4.

### 3.1 Lower Bound $\alpha_S^*(k)$

Theorem 3 below formally states the lower bound $\alpha_S^*(k)$ for the competitive ratio of any online algorithm for OSDoS.

THEOREM 3 (LOWER BOUND). *Given $\mathcal{S} = \{L, U, f\}$ for the OS-DoS problem with $k \geq 1$, no online algorithm, including those with*

*randomization, can achieve a competitive ratio smaller than $\alpha_S^*(k)$, where $\alpha_S^*(k)$ is the solution to the following equation of $\alpha$:*

$$U = \left(L - c_{\underline{k}}\right) \cdot e^{\frac{\alpha}{k} \cdot (k+1-\underline{k}-\xi)} + c_{\underline{k}} \cdot e^{\frac{\alpha}{k} \cdot (k-\underline{k})} +$$

$$c_{\underline{k}+1} \cdot \left(1 - e^{\frac{\alpha}{k}}\right) \cdot e^{\frac{\alpha}{k} \cdot (k-1-\underline{k})} + \cdots + c_k \cdot \left(1 - e^{\frac{\alpha}{k}}\right). \quad (1)$$

*In Eq. (1), $\underline{k} \in [k]$ denotes the smallest natural number such that*

$$\sum_{i=1}^{\underline{k}} (L - c_i) \geq \frac{1}{\alpha} \cdot \left(kL - \sum_{i=1}^{k} c_i\right), \quad (2)$$

*and $\xi \in (0, 1]$ denotes the unique solution to the following equation*

$$\xi = \frac{\frac{1}{\alpha} \cdot (kL - \sum_{i=1}^{k} c_i) - \sum_{i=1}^{\underline{k}-1} (L - c_i)}{L - c_{\underline{k}}}. \quad (3)$$

Theorem 3 is our main result concerning the hardness of OSDoS. To prove Theorem 3, a key step is to establish a set of necessary conditions that any $\alpha$-competitive online algorithm must satisfy. A formal proof will be provided in Section 3.3. Below, we offer several remarks to clarify the key intuitions.

- By the definition of $\underline{k}$ in Eq. (2), $\underline{k}$ represents the minimum number of units that any $\alpha$-competitive deterministic algorithm, denoted by $\text{ALG}_d$, must sell when faced with an arrival instance of $k$ identical buyers with valuation $L$, denoted by $\mathcal{I}_{iden}^{(L)} = \{L, \cdots, L\}$. Under the instance $\mathcal{I}_{iden}^{(L)}$, the maximum social welfare achievable by the offline optimal algorithm is $kL - \sum_{i=1}^{k} c_i$. Therefore, $\text{ALG}_d$ must sell at least $\underline{k}$ units to ensure $\alpha$-competitiveness, implying that $\underline{k}$ is well-defined for all values of $\alpha \geq 1$.

- Eq. (3) demonstrates that $\xi$ is defined as the fraction of the $\underline{k}$-th unit required to make Eq. (2) binding. We argue that $\xi \in (0, 1]$ is well-defined and always exists as long as there is an $\alpha$-competitive randomized algorithm, denoted as $\text{ALG}_r$. Consider running $\text{ALG}_r$ on the same instance $\mathcal{I}_{iden}^{(L)}$. In this scenario, $\text{ALG}_r$ must sell at least $\underline{k} - 1$ units plus a fraction $\xi$ of the $\underline{k}$-th unit of the item, in expectation.

- Note that, in general, a closed-form expression for the lower bound $\alpha_S^*(k)$ cannot be derived. This is expected due to the arbitrary nature of the sequence of marginal production costs. However, because of the monotonicity of $\underline{k}$, $\xi$, and the right-hand side of Eq. (1) with respect to $\alpha$, $\alpha_S^*(k)$ can be easily computed by solving Eq. (1) numerically using binary search.

In the next subsection, we construct a family of hard instances and introduce a novel representative function-based approach to derive a system of differential equations, which are crucial to proving the lower bound result in Theorem 3.

## 3.2 Representing Worst-Case Performance by (Probabilistic) Allocation Functions

### 3.2.1 *Hard Instances* $\{\mathcal{I}_v^{(\epsilon)}\}_{\forall v \in [L,U]}$. We introduce a family of hard instances based on the instance $\mathcal{I}^{(\epsilon)}$ defined as follows.

DEFINITION 1 (INSTANCE $\mathcal{I}^{(\epsilon)}$). *For an arbitrarily small $\epsilon > 0$, the instance $\mathcal{I}^{(\epsilon)}$ starts with $k$ identical buyers with valuation $L$ in the first stage, followed by a group of $k$ identical buyers per stage, with valuations continuously increasing from $L + \epsilon$ to $U$. Mathematically,*

$\mathcal{I}^{(\epsilon)}$ *is denoted as*

$$\{\underbrace{L, \ldots, L}_{k}, \underbrace{L + \epsilon, \ldots, L + \epsilon}_{k}, \underbrace{L + 2\epsilon, \ldots, L + 2\epsilon}_{k}, \ldots,$$

$$\underbrace{v, \ldots, v}_{k \text{ buyers in stage } v}, \ldots, \underbrace{U, \ldots, U}_{k}\}.$$

By Definition 1, as $\epsilon \to 0$, for any value $v \in [L, U]$, there will be $k$ buyers with valuation $v$ in $\mathcal{I}^{(\epsilon)}$, which we define as *stage-v*. For any $v \in [L, U]$, let $\mathcal{I}_v^{(\epsilon)}$ represent all the buyers in $\mathcal{I}^{(\epsilon)}$ from the beginning up to *stage-v*. For instance, if $v = L + 2\epsilon$, then $\mathcal{I}_v^{(\epsilon)}$ includes the first $3k$ buyers in $\mathcal{I}^{(\epsilon)}$ with valuations $L, L + \epsilon$, and $L + 2\epsilon$. Due to the online nature of the problem, we emphasize that $\mathcal{I}^{(\epsilon)}$ may terminate at any stage $v$. In other words, there exists a family of hard instances, $\{\mathcal{I}_v^{(\epsilon)}\}_{\forall v \in [L,U]}$, induced by $\mathcal{I}^{(\epsilon)}$. Here, $\mathcal{I}_v^{(\epsilon)}$ denotes the arrival instance of $\mathcal{I}^{(\epsilon)}$ that terminates at stage-$v$. Henceforth, we will use "*instance $\mathcal{I}_v^{(\epsilon)}$*" and "*instance $\mathcal{I}^{(\epsilon)}$ by the end of stage-v*" interchangeably.

Given any $\alpha$-competitive algorithm ALG, an arbitrary instance from $\{\mathcal{I}_v^{(\epsilon)}\}_{\forall v \in [L,U]}$ may be the one that ALG processes. Thus, for any $v \in [L, U]$, by the end of stage-$v$ of $\mathcal{I}^{(\epsilon)}$, ALG must achieve at least a $1/\alpha$ fraction of the optimal social welfare, $kv - \sum_{i=1}^{k} c_i$, which is attained by rejecting all previous buyers except for the last $k$ buyers with valuation $v$. Consequently, an $\alpha$-competitive algorithm must ensure

$$\text{ALG}\left(\mathcal{I}_v^{(\epsilon)}\right) \geq \frac{1}{\alpha} \cdot \left(kv - \sum_{i=1}^{k} c_i\right), \quad \forall v \in [L, U], \quad (4)$$

where $\text{ALG}(\mathcal{I}_v^{(\epsilon)})$ denotes the *expected* performance of ALG under the instance $\mathcal{I}_v^{(\epsilon)}$.

### 3.2.2 *Representing $\text{ALG}(\mathcal{I}_v^{(\epsilon)})$ by Allocation Functions.* For any randomized algorithm, we define $k+1$ distinct states, $\{q_i\}_{\forall i \in \{0, \cdots, k\}}$, which represent the allocation behavior of the online algorithm at any stage of instance $\mathcal{I}^{(\epsilon)}$, as follows:

- State $q_0$ corresponds to the situation where the online algorithm has not allocated any units.
- For all $i \in [k]$, state $q_i$ represents that the online algorithm has allocated *at least $i$* units of the item.

For all $v \in [L, U]$ and $i \in \{0, \cdots, k\}$, we define $\Psi_i(v) : [L, U] \to \{0, 1\}$ such that $\Psi_i(v) = 1$ if the algorithm is in state $q_i$ after processing all the buyers in $\mathcal{I}_v^{(\epsilon)}$, and $\Psi_i(v) = 0$ otherwise. Specifically, $\Psi_i(v) = 1$ if the online algorithm allocates at least $i$ units of the item at the end of stage $v$ in $\mathcal{I}^{(\epsilon)}$, which occurs with some probability depending on the algorithm's randomness. Since the instance $\mathcal{I}^{(\epsilon)}$ is deterministically defined, $\Psi_i(v)$ is a binary random variable whose distribution depends solely on the algorithm's randomness. This leads to the definition of $\boldsymbol{\psi} = \{\psi_i\}_{\forall i \in [k]}$ below.

DEFINITION 2 (ALLOCATION FUNCTIONS). *For any randomized online algorithm, let $\boldsymbol{\psi} = \{\psi_i\}_{\forall i \in [k]}$ and $\psi_i : [L, U] \to [0, 1]$ represent the functions where $\psi_i(v) = \mathbb{E}[\Psi_i(v)]$, with the expectation taken over the randomness of the algorithm.*

Based on the definition above, we have $\psi_i(v) = \Pr(\Psi_i(v) = 1)$, where $\Psi_i(v) = 1$ indicates that the algorithm is in state $q_i$ (i.e., at

least $i$ units of the item have been allocated) after processing all buyers in $\mathcal{I}_v^{(\epsilon)}$ (i.e., by the end of stage $v$ of instance $\mathcal{I}^{(\epsilon)}$). In this context, $\psi_i(v)$ represents the probability that the online algorithm has allocated at least $i$ units of the item by the end of stage $v$ in instance $\mathcal{I}^{(\epsilon)}$. Therefore, the term *probabilistic allocation functions* is used or simply *allocation functions* for brevity. The following lemma demonstrates several important properties of $\psi_i(v)$ for all $i \in [k]$ and $v \in [L, U]$.

LEMMA 1 (MONOTONICITY; CONTINUITY). *For any randomized online algorithm, $\psi_i(v) \geq \psi_{i+1}(v)$ holds for all $i \in [k]$ and $v \in [L, U]$. In addition, for any $i \in [k]$, if the allocation function $\psi_i(v)$ is discontinuous at some point $v \in [L, U]$, then there exists another algorithm $\widehat{ALG}$ such that $\hat{\psi}_i(v)$ is continuous at $v \in [L, U]$ and $\widehat{ALG}$ performs no worse than ALG.*

The proof of the above lemma is given in Appendix A.

Lemma 1 implies that it suffices to focus on randomized algorithms whose allocation functions are from the following set

$$\Omega = \Big\{ \boldsymbol{\psi} \Big| \psi_i(v) \in [0, 1], \psi_i(v) \geq \psi_{i+1}(v),$$
$$\psi_i(v) < \psi_i(v'), \forall i \in [k], v, v' \in [L, U], \text{ and } v < v' \Big\}.$$

Next, we analyze how the allocation level of an $\alpha$-competitive algorithm should evolve as new buyers with higher valuations arrive in $\mathcal{I}^{(\epsilon)}$. We argue that the expected performance of any online algorithm under the instance $\mathcal{I}^{(\epsilon)}$ can be fully represented by the $k$ allocation functions $\{\psi_i(v)\}_{\forall i \in [k]}$. Let $\text{ALG}(\mathcal{I}_v^{(\epsilon)})$ denote the expected objective value of the algorithm under instance $\mathcal{I}_v^{(\epsilon)}$. Then $\text{ALG}(\mathcal{I}_v^{(\epsilon)})$ can be framed using $\boldsymbol{\psi} = \{\psi_i(v)\}_{\forall i \in [k]}$ as follows.

PROPOSITION 1 (REPRESENTATION BASED ON $\boldsymbol{\psi}$). *For any randomized algorithm ALG under the family of hard instances $\{\mathcal{I}_v^{(\epsilon)}\}_{\forall v \in [L,U]}$ with $\epsilon \to 0$, its expected performance can be represented by its allocation functions $\{\psi_i(v)\}_{\forall i \in [k]} \in \Omega$ as follows:*

$$\begin{cases} \text{ALG}\left(\mathcal{I}_L^{(\epsilon)}\right) = \sum_{i=1}^{k} \psi_i(L) \cdot (L - c_i), \\ \text{ALG}\left(\mathcal{I}_v^{(\epsilon)}\right) = \text{ALG}\left(\mathcal{I}_L^{(\epsilon)}\right) + \sum_{i=1}^{k} \int_{\eta=L}^{v} (\eta - c_i) d\psi_i(\eta), \ \forall v \in (L, U]. \end{cases}$$

The above proposition relates the expected performance of an online algorithm to the set of allocation functions $\{\psi_i(v)\}_{\forall i \in [k]}$ that capture its dynamics under the family of hard instances $\{\mathcal{I}_v^{(\epsilon)}\}_{\forall v \in [L,U]}$. The detailed proof can be found in Appendix B.

Combining Proposition 1 and Eq. (4) gives the corollary below.

COROLLARY 4 (NECESSARY CONDITIONS). *If there exists an $\alpha$-competitive algorithm for OSDoS, then there exists $k$ allocation functions $\{\psi_i\}_{\forall i \in [k]} \in \Omega$ that satisfy the following equation:*

$$\sum_{i=1}^{k} \psi_i(L) \cdot (L - c_i) + \sum_{i=1}^{k} \int_{\eta=L}^{v} (\eta - c_i) d\psi_i(\eta)$$
$$\geq \frac{1}{\alpha} \cdot \left( kv - \sum_{i=1}^{k} c_i \right), \quad \forall v \in [L, U]. \tag{5}$$

The corollary above provides a set of necessary conditions for the allocation functions $\{\psi_i\}_{\forall i \in [k]}$ induced by any $\alpha$-competitive

algorithm. Therefore, determining a tight lower bound for OSDoS is equivalent to finding the lowest $\alpha$ such that there exists a set of allocation functions in $\Omega$ that satisfy Eq. (5).

## 3.3 Proof of Theorem 3

We now move on to prove Theorem 3. Based on the necessary conditions in Corollary 4, the lower bound can be defined as

$$\alpha_S^*(k) = \inf \Big\{ \alpha \geq 1 \Big| \text{there exist a set of } k \text{ allocation}$$
$$\text{functions } \{\psi_i(v)\}_{\forall i \in [k]} \in \Omega \text{ that satisfy Eq. (5)} \Big\}.$$

Next, we show that it is possible to find a tight design of $\{\psi_i\}_{\forall i \in [k]}$ that satisfies the necessary conditions in Eq. (5) by equality, ultimately leading to Eq. (1) in Theorem 3.

For any $\alpha \geq \alpha_S^*(k)$, let $\Gamma^{(\alpha)}$ denote the superset of the set of functions $\{\psi_i\}_{\forall i \in [k]} \in \Omega$ that satisfy Eq. (5). Note that $\Gamma^{(\alpha)} \subset \Omega$ holds for all $\alpha \geq \alpha_S^*(k)$. Let function $\chi^{(\alpha)}(v) : [L, U] \to [0, k]$ be defined as

$$\chi^{(\alpha)}(v) = \inf \left\{ \sum_{i=1}^{k} \psi_i(v) \Big| \{\psi_i(v)\}_{\forall i \in [k]} \in \Gamma^{(\alpha)} \right\}. \tag{6}$$

Based on the definition of $\chi^{(\alpha)}$, we construct a set of allocation functions $\{\psi_i^{(\alpha)}(v)\}_{\forall i \in [k]}$ as follows:

$$\psi_i^{(\alpha)}(v) = \left( \chi^{(\alpha)}(v) - (i - 1) \right) \cdot \mathbb{1}_{\{i-1 \leq \chi^{(\alpha)}(v) \leq i\}} +$$
$$\mathbb{1}_{\{\chi^{(\alpha)}(v) > i\}}, \forall v \in [L, U], \quad \forall i \in [k], \tag{7}$$

where $\mathbb{1}_{\{A\}}$ is the standard indicator function, equal to 1 if $A$ is true and 0 otherwise. In the following lemma, we argue that the set of functions $\{\psi_i^{(\alpha)}(v)\}_{\forall i \in [k]}$ is a feasible solution to Eq. (5) and satisfies it as an equality.

LEMMA 2. *For any $\alpha \geq \alpha_S^*(k)$, the functions $\{\psi_i^{(\alpha)}\}_{\forall i \in [k]}$ satisfy Eq. (5) as an equality.*

The detailed proof for the above lemma is in Appendix C. Following the definition of $\{\psi_i^{(\alpha)}(v)\}_{\forall i \in [k]}$, we observe that these functions exhibit the following property:

LEMMA 3. *For any $i \in [k]$ and $v \in [L, U]$, if $\psi_i^{(\alpha)}(v) \in (0, 1)$ holds, then $\psi_j^{(\alpha)}(v) = 1$ for all $j = 1, \cdots, i - 1$ and $\psi_j^{(\alpha)}(v) = 0$ for all $j = i + 1, \cdots, k$.*

The lemma above asserts that if the online algorithm inducing $\{\psi^{(\alpha)}\}_{\forall i}$ begins allocating unit $i$ with some positive probability to buyers in stage-$v$ of $\mathcal{I}^{(\epsilon)}$, then the algorithm must have already allocated all units $j < i$ with probability one to buyers arriving at or before stage-$v$ of $\mathcal{I}^{(\epsilon)}$. Furthermore, if the algorithm has not allocated unit $i$ with probability one by the end of stage-$v$, then all units $j > i$ remain in the system with probability one at the end of stage-$v$. Given that the marginal cost for each additional unit of resource increases, the algorithm should only produce and allocate a new unit once all previously produced units have been fully allocated.

According to Lemma 2, the inequality in Eq. (5) can be replaced with an equality. By combining Lemma 2 with Lemma 3, we conclude that there exists a unique set of functions that satisfy Eq.

(5) as an equality and also fulfill the property stated in Lemma 3. Proposition 2 below formally states this result.

PROPOSITION 2. *For any* $\alpha \geq \alpha^*_S(k)$, *there exist a unique set of* $k$ *allocation functions* $\{\psi_i^{(\alpha)}\}_{\forall i \in [k]} \in \Omega$ *that satisfy Eq.* (5) *by equality:*

$$\psi_i^{(\alpha)}(v) = 1, \quad i = 1, \ldots, \underline{k} - 1,$$

$$\psi_{\underline{k}}^{(\alpha)}(v) = \begin{cases} \xi + \frac{k}{\alpha} \cdot \ln\left(\frac{v - c_{\underline{k}}}{L - c_{\underline{k}}}\right) & v \in [L, u_{\underline{k}}], \\ 1 & v > u_{\underline{k}}, \end{cases}$$

$$\psi_i^{(\alpha)}(v) = \begin{cases} 0 & v \leq \ell_i, \\ \frac{k}{\alpha} \cdot \ln\left(\frac{v - c_i}{\ell_i - c_i}\right) & v \in [\ell_i, u_i], \quad i = \underline{k} + 1, \ldots, k - 1, \\ 1 & v \geq u_i, \end{cases}$$

$$\psi_k^{(\alpha)}(v) = \begin{cases} 0 & v \leq \ell_k, \\ \frac{k}{\alpha} \cdot \ln\left(\frac{v - c_k}{\ell_k - c_k}\right) & v \in [\ell_k, U], \end{cases}$$

*where the intervals* $\{[\ell_i, u_i]\}_{\forall i}$ *are specified by*

$$u_{\underline{k}} = \ell_{\underline{k}+1} = (L - c_{\underline{k}}) \cdot e^{(1-\xi) \cdot \frac{\alpha}{k}} + c_{\underline{k}}, \tag{8}$$

$$u_i = \ell_{i+1} = (\ell_i - c_i) \cdot e^{\alpha/k} + c_i \quad \forall i = \underline{k} + 1, \ldots, k. \tag{9}$$

Recall that the parameters $\underline{k}$ and $\xi$ are defined in Eq. (2) and Eq. (3), respectively. Once $\alpha$ is given, both $\underline{k}$ and $\xi$ can be uniquely determined. Therefore, the set of allocation functions $\{\psi_i^{(\alpha)}\}_{\forall i \in [k]}$ given in Proposition 2 can also be explicitly computed once $\alpha$ is given. The full proof of how to derive the explicit designs of $\{\psi_i^{(\alpha)}\}_{\forall i \in [k]}$ is given in Appendix D.

Putting together Eq. (8) and Eq. (9), we have

$$u_k = (L - c_{\underline{k}}) \cdot e^{\frac{\alpha}{k} \cdot (k+1-\underline{k}-\xi)} + c_{\underline{k}} \cdot e^{\frac{\alpha}{k} \cdot (k-\underline{k})} +$$

$$c_{\underline{k}+1} \cdot (1 - e^{\frac{\alpha}{k}}) \cdot e^{\frac{\alpha}{k} \cdot (k-1-\underline{k})} + \cdots + c_k \cdot (1 - e^{\frac{\alpha}{k}}).$$

Note that the right-hand side of the equation above is increasing in $\alpha$. Therefore, as $\alpha$ decreases, the value of $u_k$ also decreases and will eventually fall below $U$ for a specific value of $\alpha$. Consequently, according to the definition of $\psi_k^{(\alpha)}$ in Proposition 2, $\psi_k^{(\alpha)}(U)$ will exceed 1 (since $\psi_k^{(\alpha)}(U) > \psi_k^{(\alpha)}(u_k)$, and based on Eq. (9), $\psi_k^{(\alpha)}(u_k)$ is equal to one). However, this will generate an infeasible allocation function $\psi_k^{(\alpha)}$, as we require that $\psi_k^{(\alpha)}(v) \leq 1$ holds for all $v \in [L, U]$. As a result, for those values of $\alpha$ where $u_k < U$, the set of $k$ allocation functions $\{\psi_i^{(\alpha)}\}_{\forall i \in [k]}$ obtained in Proposition 2 becomes infeasible, meaning that $\alpha$ must be less than $\alpha^*_S(k)$. Therefore, the lower bound $\alpha^*_S(k)$ is the value of $\alpha$ for which $u_k = U$, and this gives Eq. (1) in Theorem 3. Thus, we complete the proof of Theorem 3.

## 4 r-Dynamic: A Randomized Dynamic Posted Price Mechanisms

We propose a randomized dynamic pricing mechanism (r-Dynamic), as described in Algorithm 1, to solve the OSDoS problem. Before the buyers arrive, r-Dynamic samples $k$ independent random prices $\{P_i\}_{\forall i \in [k]}$, where $P_i$ is the price for the $i$-th unit of the item. Specifically, for each unit $i \in [k]$, a random seed $s_i$ is drawn from the uniform distribution Unif$(0, 1)$, and the random price is set

---

**Algorithm 1** Randomized Dynamic Pricing (r-Dynamic) for OSDoS

1: **Input:** pricing functions $\{\phi_i\}_{\forall i \in [k]}$;
2: **Initiate:** index of the unit to be sold $\kappa_1 = 1$;
3: Generate a random seed vector $\mathbf{s} = \{s_i\}_{\forall i \in [k]}$, each element sampled independently from uniform distribution Unif$(0, 1)$;
4: Set a price vector $\mathbf{P} = \{P_i\}_{\forall i \in [k]}$, where $P_i = \phi_i(s_i)$;
5: **while** buyer $t$ arrives **do**
6:     **if** $\kappa_t \leq k$ **then**:
7:         Post the price $p_t = P_{\kappa_t}$ to buyer $t$;
8:         **if** buyer $t$ accepts the price **then**
9:             One unit is sold and set $x_t = 1$;
10:         **end if**
11:     **end if**
12:     Update $\kappa_{t+1} = \kappa_t + x_t$.     ▷ $x_t = 0$ if buyer $t$ declines $p_t$.
13: **end while**

---

as $P_i = \phi_i(s_i)$, where $\phi_i(s_i)$ is the *pricing function* designed for the $i$-th unit. r-Dynamic then posts the price of the available unit with the smallest index from $\{P_i\}_{\forall i \in [k]}$ to the online arriving buyers.

For all $i \in [k]$, the pricing function $\phi_i : [0, 1] \rightarrow [L_i, U_i]$ is constructed such that the $k$ price intervals $\{[L_i, U_i]\}_{\forall i \in [k]}$ span the entire range of $[L, U]$, where $L = L_1 \leq U_1 = L_2 \leq U_2 \leq \cdots \leq U_{k-1} = L_k \leq U_k = U$. That is, the upper boundary of $\phi_i$ (i.e., the maximum price of $P_i$) is the lower boundary of $\phi_{i+1}$ (i.e., the minimum price of $P_{i+1}$). As a result, the posted prices will always be non-decreasing (i.e., $P_1 \leq P_2 \leq \cdots \leq P_k$), regardless of the realization of the random seeds $\{s_i\}_{\forall i \in [k]}$. This design ensures that units with higher production costs are sold at higher prices, which is consistent with the natural pricing scheme where more expensive units reflect higher production costs.

### 4.1 Asymptotic Optimality of r-Dynamic

We show that by carefully designing the pricing functions, r-Dynamic achieves an asymptotically optimal competitive ratio.

THEOREM 5. *Given* $S = \{L, U, f\}$ *for the OSDoS problem with* $k \geq 1$, *r-Dynamic is* $\alpha^*_S(k) \cdot \exp(\frac{\alpha^*_S(k)}{k})$-*competitive when the pricing functions are given by*

$$\phi_i(s) = L, \quad \forall s \in [0, 1], i \in [\underline{k}^* - 1],$$

$$\phi_{\underline{k}^*}(s) = \begin{cases} L & s \in [0, \xi^*], \\ (L - c_{\underline{k}^*}) \cdot e^{(s - \xi^*) \cdot \alpha^*_S(k)/k} + c_{\underline{k}^*} & s \in [\xi^*, 1], \end{cases}$$

$$\phi_i(s) = (L_i - c_i) \cdot e^{s \cdot \alpha^*_S(k)/k} + c_i, \quad \forall s \in [0, 1], i = \underline{k}^* + 1, \ldots, k,$$

*where* $\underline{k}^*$ *and* $\xi^*$ *are respectively the values of* $\underline{k}$ *and* $\xi$ *defined in Theorem 3, corresponding to* $\alpha = \alpha^*_S(k)$, *and the price intervals* $\{[L_i, U_i]\}_{\forall i \in [k]}$ *are given as follows:*

$$U_{\underline{k}^*} = L_{\underline{k}^*+1} = (L - c_{\underline{k}^*}) \cdot e^{(1-\xi^*) \cdot \alpha^*_S(k)/k} + c_{\underline{k}^*}, \tag{10}$$

$$U_i = L_{i+1} = (L_i - c_i) \cdot e^{\alpha^*_S(k)/k} + c_i, \quad \forall i = \underline{k}^* + 1, \ldots, k. \tag{11}$$

We provide a proof sketch of Theorem 5 in Section 4.3. At a high level, the design of the pricing functions $\{\phi_i(s)\}_{\forall i \in [k]}$ is inspired by the dynamics of an $\alpha^*_S(k)$-competitive algorithm on the arrival instance $\mathcal{I}^{(\epsilon)}$ studied in the lower bound section. Essentially, the inverse of the pricing function $\phi_i(s)$, defined as $\phi_i^{-1}(v) = \sup\{s :$

$\phi_i(s) \leq v\}$, follows the same design as $\psi_i^{(\alpha)}(v)$ in Proposition 2 when $\alpha = \alpha_{\mathcal{S}}^*(k)$, namely, $\psi_i^{(\alpha_{\mathcal{S}}^*(k))}(v) = \sup\{s : \phi_i(s) \leq v\}$.

**Asymptotic optimality of r-Dynamic in general settings.** It is important to note that previous studies (e.g., [12, 23]) have shown that $\alpha_{\mathcal{S}}^*(k)$ remains bounded by a constant as $k$ approaches infinity.[1] Thus, the competitive ratio of r-Dynamic approaches the lower bound $\alpha_{\mathcal{S}}^*(k)$ as $k$ goes to infinity (i.e., $\alpha_{\mathcal{S}}^*(k)/k \to 0$ as $k \to \infty$), meaning that r-Dynamic is asymptotically optimal.

**Exact optimality of r-Dynamic when $k = 2$.** For the small inventory case of $k = 2$, a tighter analysis shows that r-Dynamic is $\alpha_{\mathcal{S}}^*(2)$-competitive using the same design of pricing functions in Theorem 5, where $\alpha_{\mathcal{S}}^*(2)$ is the lower bound obtained in Theorem 3 for $k = 2$. This indicates that r-Dynamic is not just asymptotically optimal, but also optimal in the small inventory setting when $k = 2$. The corollary below formalizes this result.

COROLLARY 6. *Given $\mathcal{S} = \{L, U, f\}$ for the OSDoS problem with $k = 2$, r-Dynamic is $\alpha_{\mathcal{S}}^*(2)$-competitive when $\phi_1 : [0, 1] \to [L_1, U_1]$ and $\phi_2 : [0, 1] \to [L_2, U_2]$ are designed as follows:*

- *If $\alpha_{\mathcal{S}}^*(2) \geq \frac{2L - c_1 - c_2}{L - c_1}$, then:*

$$\phi_1(s) = \begin{cases} L & s \in [0, \xi^*], \\ (L - c_1) \cdot e^{(s - \xi^*) \cdot \alpha_{\mathcal{S}}^*(2)/2} + c_1 & s \in [\xi^*, 1], \end{cases}$$

$$\phi_2(s) = (L_2 - c_2) \cdot e^{s \cdot \alpha_{\mathcal{S}}^*(2)/2} + c_2 \qquad \forall s \in [0, 1].$$

*In this case, the price intervals and $\xi^*$ are given by*

$$L_1 = L, U_1 = L_2 = (L - c_1) \cdot e^{(1 - \xi^*) \cdot \alpha_{\mathcal{S}}^*(2)/2} + c_1, U_2 = U,$$

$$\xi^* = \frac{1}{\alpha_{\mathcal{S}}^*(2)} \cdot \frac{(2L - c_1 - c_2)}{L - c_1}.$$

- *If $\alpha_{\mathcal{S}}^*(2) < \frac{2L - c_1 - c_2}{L - c_1}$, then:*

$$\phi_1(s) = L, \qquad \forall s \in [0, 1],$$

$$\phi_2(s) = \begin{cases} L & s \in [0, \xi^*], \\ (L - c_2) \cdot e^{(s - \xi^*) \cdot \alpha_{\mathcal{S}}^*(2)/2} + c_2 & s \in [\xi^*, 1]. \end{cases}$$

*In this case, the price intervals and $\xi^*$ are given by*

$$L_1 = U_1 = L_2 = L, \qquad U_2 = U,$$

$$\xi^* = \frac{(2L - c_1 - c_2)/\alpha_{\mathcal{S}}^*(2) - (L - c_1)}{L - c_2}.$$

The proof of the corollary above is given in Appendix F. In the following two subsections, we first evaluate the empirical performance of r-Dynamic and then provide a proof sketch of Theorem 5 to show the asymptotic optimality of r-Dynamic.

## 4.2 Empirical Performance of r-Dynamic

We perform three experiments to evaluate the empirical performance of r-Dynamic and compare its performance to two other algorithms, d-Dynamic [17] and r-Static [20]. Throughout the three experiments, the setup $\mathcal{S}$ is fixed to be $\{L = 1, U = 30, f(i) = i^2/16\}$

---

[1]Recall that all buyers' valuations are bounded within $[L, U]$. Thus, any naive algorithm will be at least $\frac{kU - f(k)}{kL - f(k)}$-competitive by allocating $k$ units to buyers with the lowest valuation, $L$, while the offline optimal algorithm allocates all $k$ units to buyers with the highest valuation, $U$.

and $k = 10$. To stimulate different arrival patterns of buyers, we consider the following three types of instances:

- **Instance-IID**: We generate the valuations of 1000 buyers using the truncated normal distribution $N(15, 15)_{[1,30]}$.
- **Instance-Sorted**: We generate 1000 buyers using the same approach as **Instance-IID**, and then sort these buyers in increasing order by their valuations. This instance mimics the hard instance $\mathcal{I}^{(\epsilon)}$.
- **Instance-Low2High**: We generate the valuations of 500 buyers using truncated normal distribution $N(7.5, 7.5)_{[1,30]}$. Following these 500 buyers, we generate another 500 buyers using distribution $N(22.5, 7.5)_{[1,30]}$.

Figure 2 presents the CDF plot of the empirical competitive ratios for the three algorithms r-Dynamic, d-Dynamic, and r-Static, evaluated on 300 instances from each type of instance. In Figure 2(a), r-Dynamic significantly outperforms the other two algorithms under **Instance-Sorted**. This is because the valuations of online arrivals are increasing, similar to the hard instance $\mathcal{I}^{(\epsilon)}$ defined in Section 3.2. This result confirms the superior performance of r-Dynamic under difficult instances compared to the other algorithms. Additionally, Figure 2(a) demonstrates that r-Dynamic's performance is very close to the lower bound $\alpha_{\mathcal{S}}^*(10)$, suggesting that r-Dynamic may not only be asymptotically optimal in the large $k$ regime but also near-optimal in the small $k$ regime. In Figure 2(b), **Instance-Low2High** consists of two phases: low-valued buyers arriving first, followed by high-valued buyers. This instance is simpler than **Instance-Sorted**, and the performance of all three algorithms improves, with r-Dynamic continuing to outperform the others. Finally, in Figure 2(c), under **Instance-IID**, all algorithms achieve a competitive ratio close to 1, with r-Dynamic and d-Dynamic performing similarly. From these figures, we observe that the advantage of r-Dynamic becomes increasingly evident as the instances become more difficult, particularly when low-valued buyers arrive before high-valued buyers.

## 4.3 Proof Sketch of Theorem 5

For an arbitrary arrival instance $\mathcal{I} = \{v_t\}_{\forall t \in [T]}$, we prove that r-Dynamic is $\alpha_{\mathcal{S}}^*(k) \cdot \exp(\frac{\alpha_{\mathcal{S}}^*(k)}{k})$-competitive if the pricing functions $\{\phi_i\}_{\forall i \in [k]}$ are designed according to Theorem 5.

Recall that the random price vector $\mathbf{P} = \{P_i\}_{i \in [k]}$ is generated using the pricing functions $\{\phi_i\}_{\forall i \in [k]}$ at the start of r-Dynamic (line 3 of Algorithm 1). Hereafter, we will refer to Algorithm 1 as r-Dynamic($\mathbf{P}$) to indicate that the algorithm is executed with the random price vector $\mathbf{P}$. Based on the design of $\{\phi_i\}_{\forall i \in [k]}$ in Theorem 5, the first $\underline{k}^* - 1$ prices in $\mathbf{P}$ are all $L$'s (i.e., $P_1 = \cdots = P_{\underline{k}^* - 1} = L$), the $\underline{k}^*$-th price $P_{\underline{k}^*}$ is a random variable within $[L, U_{\underline{k}^*}]$, and for all $i \in \{\underline{k}^* + 1, \cdots, k\}$, the $i$-th price $P_i$ is a random variable within $[L_i, U_i]$. Here, the values of $\underline{k}^*$ and $\{[L_i, U_i]\}_{\forall i}$ are all defined in Theorem 5.

Let $\mathcal{P}$ denote the support of all possible values of the random price vector $\mathbf{P}$:

$$\mathcal{P} = \{L\}^{\underline{k}^* - 1} \times [L, U_{\underline{k}^*}] \times \prod_{i \in \{\underline{k}^* + 1, \cdots, k\}} [L_i, U_i].$$

Given a price vector $\mathbf{P} \in \mathcal{P}$, let $W(\mathbf{P})$ represent the total number of items allocated by r-Dynamic($\mathbf{P}$) under the input instance $\mathcal{I}$. Since

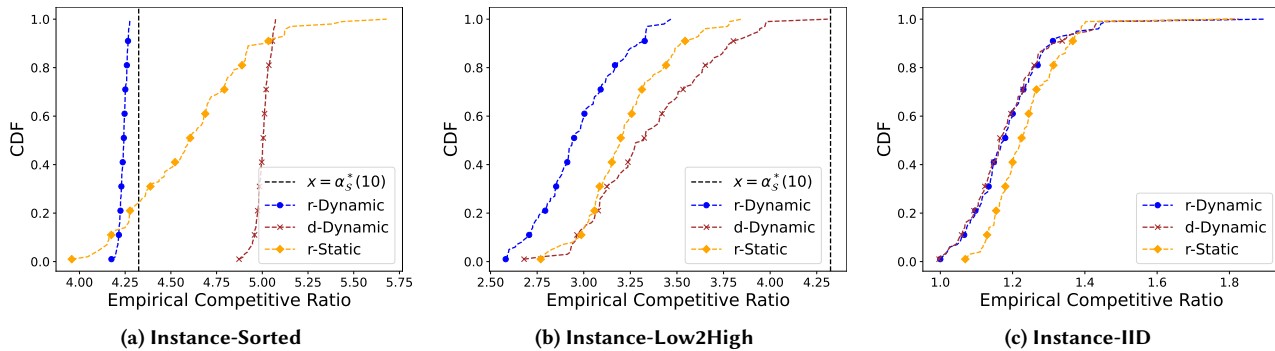

**Figure 2: CDF plots of empirical competitive ratios of r-Dynamic (Algorithm 1), d-Dynamic [23] and r-Static [20].**

$\mathbf{P}$ is a random variable, $W(\mathbf{P})$ is also a random variable. For clarity, we will sometimes omit the price vector and refer to it simply as $W$ whenever the context is clear.

Let $\omega$ be the maximum value in the support of the random variable $W$ (i.e., $\omega$ is the maximum possible value of $W(\mathbf{P})$ for all $\mathbf{P} \in \mathcal{P}$). Thus, $\omega$ is a deterministic value that depends only on the input instance $\mathcal{I}$. In addition, let $\boldsymbol{\pi} \in \mathcal{P}$ be a price vector such that r-Dynamic($\boldsymbol{\pi}$) allocates the $\omega$-th item earlier than any other price vector in the set $\mathcal{P}$. That is, for all $\mathbf{P} \in \mathcal{P}$, r-Dynamic($\mathbf{P}$) allocates the $\omega$-th item no earlier than that of r-Dynamic($\boldsymbol{\pi}$). Let us define the set $\{\tau_i\}_{\forall i \in [\omega]}$ so that $\tau_i$ is the arrival time of the buyer in the instance $\mathcal{I}$ to whom r-Dynamic($\boldsymbol{\pi}$) allocates the $i$-th unit. Note that for all $i \in \{1, \cdots, \omega\}$, $\tau_i$ is a deterministic value once $\boldsymbol{\pi}$ and $\mathcal{I}$ are given. Let the random variable $W^{\tau_\omega}(\mathbf{P})$ denote the total number of items allocated by r-Dynamic($\mathbf{P}$) after the arrival of buyer $\tau_\omega$ in the instance $\mathcal{I}$. The lemma below shows that the random variable $W^{\tau_\omega}(\mathbf{P})$ is always lower bounded by $\omega - 1$.

LEMMA 4. *Given instance $\mathcal{I}$, $W^{\tau_\omega}(\mathbf{P}) \geq \omega - 1$ holds for all $\mathbf{P} \in \mathcal{P}$.*

Lemma 4 greatly simplifies the analysis of r-Dynamic since it implies that the support of the random variable $W^{\tau_\omega}$ consists only of two values: $\omega - 1$ and $\omega$ (note that all $W$'s are upper bounded by $\omega$). The intuition behind Lemma 4 is as follows. For all $i \in \{1, \cdots, \omega\}$, recall that $\tau_i$ denotes the arrival time of the buyer in the instance $\mathcal{I}$ who receives the $i$-th unit under r-Dynamic($\boldsymbol{\pi}$). Upon the arrival of buyer $\tau_i$, if the number of items allocated by r-Dynamic($\mathbf{P}$) is less than $i - 1$, then the current $\tau_i$-th buyer will definitely accept the price offered to her, ensuring that one more unit will be sold. As a result, at least $\omega - 1$ items will be allocated by the end of time $\tau_\omega$. Lemma 4 thus follows.

The following two lemmas help us lower bound the expected performance of r-Dynamic on input instance $\mathcal{I}$ and upper bound the objective of the offline optimal algorithm, respectively.

LEMMA 5. *If a buyer in instance $\mathcal{I}$ arrives before time $\tau_\omega$ with a valuation within $[L_\omega, U]$, then for all $\mathbf{P} \in \mathcal{P}$, r-Dynamic($\mathbf{P}$) will allocate one unit of the item to that buyer.*

Lemma 5 can be proved as follows. By definition, $\tau_\omega$ is the earliest time across all possible price vectors in $\mathcal{P}$ that the production level exceeds $\omega - 1$, causing the posted price to exceed $U_{\omega-1}$. Thus, for all possible realization of $\mathbf{P} \in \mathcal{P}$, the posted prices by r-Dynamic remain below $U_{\omega-1}$ before the arrival of buyer at time $\tau_\omega$. Consequently, when a buyer with a valuation within $[L_\omega, U]$ arrives

before time $\tau_\omega$, the buyer accepts the price posted to him (since $L_\omega \geq U_{\omega-1}$) and a unit of item will thus be allocated to this buyer.

LEMMA 6. *There are no buyers in instance $\mathcal{I}$ with a valuation within $[U_\omega, U]$ arriving after time $\tau_\omega$, namely, the valuations of all buyers arrive after $\tau_\omega$ are less than $U_\omega$.*

y The above lemma can be proved by contradiction. If there exists a buyer arriving after time $\tau_\omega$ with a valuation within $[U_\omega, U]$, then there must exist a price vector in $\mathcal{P}$, say $\mathbf{P}'$, such that r-Dynamic($\mathbf{P}'$) will allocate more than $\omega$ units, contradicting the definition of $\omega$.

Applying Lemma 5 and observing that r-Dynamic sells at least $\omega - 1$ units, we can derive a lower bound on the expected performance of r-Dynamic. Conversely, using the lemma 6 and the fact that for all $\mathbf{P} \in \mathcal{P}$, the allocation level of r-Dynamic never exceeds $\omega$, we can upper bound the objective of the offline optimal algorithm. The combination of these two bounds yields the final competitive ratio of r-Dynamic. For the full proof of Theorem 5, refer to Appendix E.

## 5 Conclusions and Future Work

In this paper, we studied online $k$-selection with production costs that exhibit diseconomies of scale (OSDoS) and developed novel randomized dynamic pricing mechanisms with the best-known competitive ratios. Specifically, our randomized dynamic pricing scheme provides tight guarantees in both the small and large inventory settings (i.e., small and large $k$), addressing the gap left by [23]. These findings advance the theoretical understanding of OS-DoS and offer practical insights for designing randomized dynamic pricing mechanisms in online resource allocation problems with increasing marginal production costs.

This work highlights several promising directions for future research. First, we conjecture that our proposed randomized pricing mechanism is optimal for all $k \geq 1$. However, a more refined analysis is required to establish or refute its optimality for $k \geq 3$. Additionally, extending our results to multi-resource or combinatorial settings could reveal new insights into online resource allocation with diseconomies of scale in more complex environments. Furthermore, it would be valuable to explore other metrics, such as risk and fairness, in online allocation and selection to ensure that the developed randomized pricing mechanisms not only maximize efficiency but also promote reliable and equitable outcomes.

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

## A Proof of Lemma 1

We first prove the monotonicity. Following the definition of the state variables $q_{i+1}$ and $q_i$, when the random variable $\Psi_{i+1}(v)$ is realized to be 1, r-Dynamic must have allocated at least $i+1$ units, and thus $\Psi_i(v) = 1$ must hold. Consequently, it follows that $\psi_i(v) \geq \psi_{i+1}(v)$.

We now move on to prove the continuity of $\psi_i(v)$ in $v \in [L, U]$. Let ALG be an $\alpha$-competitive algorithm. For some $v \in (L, U)$ and $i \in [k]$, let the function $\psi_i(.)$ corresponding to ALG be non-continuous at $v$. Let $\lim_{x \to v^-} \psi_i(v) = v$ and $\psi_i(v) = \lim_{x \to v^+} \psi_i(v) = v + \delta$, for some $\delta > 0$. Then the algorithm must be selling at least in expectation a $\delta$-fraction of the $i$-th unit to the buyers with valuation $v$ in instance $\mathcal{I}$. Conversely, for ALG to be $\alpha$-competitive, the expected objective of the algorithm before the arrival of buyers with valuation $v$, $\text{ALG}(\mathcal{I}_{v^-}^{(\epsilon)})$, must be at least equal to $\frac{1}{\alpha} \cdot \text{OPT}(\mathcal{I}_{v^-}^{(\epsilon)}) = \frac{1}{\alpha} \cdot \text{OPT}(\mathcal{I}_v^{(\epsilon)})$, where $\text{OPT}(\mathcal{I}_v^{(\epsilon)})$ denotes the objective value of the offline optimal algorithm on the hard instance $\mathcal{I}^{(\epsilon)}$ up to the end of stage-$v$. It can be seen that selling in expectation at least a $\delta$ fraction of the $i$-th unit is unnecessary and ALG could save this fraction of the unit and sell it to buyers with higher valuations. In other words, we can construct another online algorithm, say $\widehat{\text{ALG}}$, that follows ALG up to the arrival of buyers with valuation $v$, but sells the $\delta$-fraction of the $i$-th unit to buyers with valuation strictly greater than $v$ instead. It is easy to see that $\widehat{\text{ALG}}$ will obtain a better objective value with its $\hat{\psi}_i$ being continuous at $v$. Lemma 1 follows by repeating the same process for any other discontinuous point of $\psi_i(v)$.

## B Proof of Proposition 1

For any randomized algorithm ALG, let $D(L)$ denote the number of units that ALG allocates under the instance $\mathcal{I}_L^{(\epsilon)}$ (i.e., the instance $\mathcal{I}^{(\epsilon)}$ by the end of stage-$L$). Thus, $D(L)$ is a random variable taking values from 0 to $k$. Based on definition of $D(L)$, $\text{ALG}(\mathcal{I}_L^{(\epsilon)})$ can be computed as follows:

$$\text{ALG}\left(\mathcal{I}_L^{(\epsilon)}\right) = \mathbb{E}\left[D(L) \cdot L - \sum_{i=1}^{D(L)} c_i\right],$$

where the expectation is taken with respect to the randomness of $D(L)$ (the distribution depends on the randomness of the algorithm ALG). Let the indicator function $\mathbb{1}_{\{D(L)=j\}} = 1$ if ALG allocates exactly $j$ units at the end of stage-$L$, and $\mathbb{1}_{\{D(L)=j\}} = 0$ otherwise. Based on definition of the random variables $\{\Psi_i(L)\}_{\forall i \in [k]}$, we argue that:

$$\mathbb{1}_{\{D(L)=j\}} = \Psi_j(L) - \Psi_{j+1}(L), \qquad 1 \leq j \leq k. \qquad (12)$$

Here, $\Psi_{k+1}(L) = 0$ always holds. To see why Eq. (12) is true, consider the case where the random variable $D(L) = j$, then:

$$\Psi_i(L) = 1, \quad \forall i \leq j,$$
$$\Psi_i(L) = 0, \quad \forall i > j.$$

From the equation above, we can observe that when the indicator function $\mathbb{1}_{\{D(L)=j\}} = 1$, $\Psi_{j+1}(L) - \Psi_j(L) = 1$ holds. For the case when $\mathbb{1}_{\{D(L)=j\}} = 0$, if $D(L) < j$, then $\Psi_j(L) = \Psi_{j+1}(L) = 0$ and $\mathbb{1}_{\{D(L)=j\}} = \Psi_j(L) - \Psi_{j+1}(L)$ follows. For the case $D(L) > j$, the two equations $\Psi_j(L) = \Psi_{j+1}(L) = 1$ and $\mathbb{1}_{\{D(L)=j\}} = \Psi_j(L) -$

$\Psi_{j+1}(L)$ again follow. As a result, $\text{ALG}(\mathcal{I}_L^{(\epsilon)})$ can be computed as follows:

$$\text{ALG}\left(\mathcal{I}_L^{(\epsilon)}\right) = \mathbb{E}\left[D(L) \cdot L - \sum_{i=1}^{D(L)} c_i\right]$$

$$= \sum_{j=1}^{k} \mathbb{E}\left[\mathbb{1}_{\{D(L)=j\}}\right] \cdot \left(j \cdot L - \sum_{i=1}^{j} c_i\right)$$

$$= \sum_{j=1}^{k} \mathbb{E}\left[\Psi_j(L) - \Psi_{j+1}(L)\right] \cdot \left(j \cdot L - \sum_{i=1}^{j} c_i\right)$$

$$= \sum_{j=1}^{k} \left(\psi_j(L) - \psi_{j+1}(L)\right) \cdot \left(j \cdot L - \sum_{i=1}^{j} c_i\right)$$

$$= \sum_{j=1}^{k} \psi_j(L) \cdot (L - c_j) - \psi_{k+1}(L) \cdot \left(k \cdot L - \sum_{i=1}^{k} c_i\right)$$

$$= \sum_{j=1}^{k} \psi_j(L) \cdot (L - c_j).$$

Now, let us compute the objective of the $\alpha$-competitive algorithm at the end of stage-$v$, $\forall v \in (L, U)$. Let the random variable $X_i(v)$ be the value obtained from allocating the $i$-th unit of the item at the end of some stage-$v \in (L, U)$. It follows that

$$\mathbb{E}[X_i(v) - c_i] = \psi_i(L) \cdot (L - c_i) +$$

$$\mathbb{E}\left[\int_{\eta=L}^{v} (\eta - c_i) \cdot (\Psi_i(\eta) - \Psi_i(\eta - d\eta))\right]$$

$$= \psi_i(L) \cdot (L - c_i) + \int_{\eta=L}^{v} (\eta - c_i) \cdot \mathbb{E}[\Psi_i(\eta) - \Psi_i(\eta - d\eta)]$$

$$= \psi_i(L) \cdot (L - c_i) + \int_{\eta=L}^{v} (\eta - c_i) \cdot \left(\psi_i(\eta) - \psi_i(\eta - d\eta)\right)$$

$$= \psi_i(L) \cdot (L - c_i) + \int_{\eta=L}^{v} (\eta - c_i) \cdot d\psi_i(\eta),$$

where the first equality follows because if the $i$-th unit is allocated at some stage $\eta$, then the algorithm must have sold at least $i$ units of the item by the end of $\eta$, leading to $\Psi(\eta) = 1$. Additionally, if the $i$-th unit is allocated at stage $\eta$, then at stage $\eta - d\eta$, the algorithm must have allocated fewer than $i$ units, indicating that $\Psi_i(\eta - d\eta) = 0$. The last equality follows because the function $\psi_i \in \Omega$ is continuous. Note that if the function $\psi_i$ is not differentiable at some value $v$, then we use the right derivative of $\psi_i$ at that point. Putting together the above results, it follows that:

$$\text{ALG}\left(\mathcal{I}_v^{(\epsilon)}\right)$$

$$= \sum_{i=1}^{k} \mathbb{E}[X_i(v) - c_i]$$

$$= \sum_{i=1}^{k} \left[\psi_i(L) \cdot (L - c_i) + \int_{\eta=L}^{v} (\eta - c_i) \cdot d\psi_i(\eta)\right],$$

$$= \text{ALG}\left(\mathcal{I}_L^{(\epsilon)}\right) + \sum_{i=1}^{k} \int_{\eta=L}^{v} (\eta - c_i) \cdot d\psi_i(\eta), \quad \forall v \in (L, U).$$

Proposition 1 thus follows.

## C  Proof of Lemma 2

For any $v \in [L, U]$, let us define $C_v$ as follows:

$$C_v = C_L + \sum_{i=1}^{k} \int_{\eta=L}^{v} (\eta - c_i) d\psi_i^{\alpha}(\eta), \quad \forall v \in (L, U]$$

$$C_L = \sum_{i=1}^{k} \psi_i^{\alpha}(L) \cdot (L - c_i).$$

To prove Lemma 2, we need to first prove the feasibility of $\{\psi_i^{(\alpha)}\}_{\forall i \in [k]}$, namely, $C_v$ is greater than $\frac{1}{\alpha} \cdot (k \cdot v - \sum_i c_i)$ for all $v \in [L, U]$.

For some $v \in [L, U]$, based on the definition of $\chi^{\alpha}(v)$ in Eq. (6), there exist a set of functions $\{\psi_i(v)\}_{\forall i \in [k]}$ that satisfy Eq. (5) and in the meanwhile, for some arbitrary small value $\epsilon$, we have:

$$\chi^{\alpha}(v) + \epsilon \geq \sum_{i=1}^{k} \psi_i(v). \tag{13}$$

Next, using integration by parts, we have

$$C_v = C_L + \sum_{i=1}^{k} \int_{\eta=L}^{v} (\eta - c_i) d\psi_i^{\alpha}(\eta)$$

$$= C_L + \sum_{i=1}^{k} \psi_i^{\alpha}(v) \cdot (v - c_i) -$$

$$\sum_{i=1}^{k} \psi_i^{\alpha}(L) \cdot (L - c_i) - \int_{\eta=L}^{v} \left(\sum_{i=1}^{k} \psi_i^{\alpha}(\eta)\right) d\eta$$

$$= \sum_{i=1}^{k} \psi_i^{\alpha}(v) \cdot (v - c_i) - \int_{\eta=L}^{v} \left(\sum_{i=1}^{k} \psi_i^{\alpha}(\eta)\right) d\eta$$

$$= v \cdot \left(\sum_{i=1}^{k} \psi_i^{\alpha}(p)\right) - \sum_{i=1}^{k} \psi_i^{\alpha}(v) \cdot c_i - \int_{\eta=L}^{v} \left(\sum_{i=1}^{k} \psi_i^{\alpha}(\eta)\right) d\eta$$

$$= v \cdot \chi^{\alpha}(v) - \sum_{i=1}^{k} \psi_i^{\alpha}(v) \cdot c_i - \int_{\eta=L}^{v} \chi^{\alpha}(v) d\eta,$$

where the last equality follows the definition of $\{\psi_i^{\alpha}\}_{\forall i \in [k]}$ in Eq. (7). Thus, we have

$$C_v = v \cdot \chi^{\alpha}(v) - \sum_{i=1}^{k} \psi_i^{\alpha}(v) \cdot c_i - \int_{\eta=L}^{v} \chi^{\alpha}(v) \cdot d\eta,$$

$$\geq v \cdot \sum_{i=1}^{k} \psi_i(v) - v \cdot \epsilon - \sum_{i=1}^{k} \psi_i^{\alpha}(v) \cdot c_i - \int_{\eta=L}^{v} \chi^{\alpha}(v) \cdot d\eta,$$

$$\geq v \cdot \sum_{i=1}^{k} \psi_i(v) - \int_{\eta=L}^{v} \left(\sum_{i=1}^{k} \psi_i(\eta)\right) \cdot d\eta$$

$$- \sum_{i=1}^{k} \psi_i^{\alpha}(v) \cdot c_i - v \cdot \epsilon, \tag{14}$$

where the first inequality follows Eq. (13) and the second inequality directly follows the definition of $\chi^{\alpha}(v)$ (recall that $\chi^{\alpha}(v) \leq \sum_{i=1}^{k} \psi_i(v)$ holds for all $v \in [L, U]$).

By the definition of $\{\psi_i^{\alpha}\}_{\forall i \in [k]}$, we have $\sum_{i \in [k]} \psi_i^{\alpha}(v) = \chi^{\alpha}(v)$. Putting together the inequality $\chi^{\alpha}(v) \leq \sum_{i=1}^{k} \psi_i(v)$ and the fact

that productions costs are increasing, we have

$$\sum_{i=1}^{k} \psi_i^\alpha(v) \cdot c_i \leq \sum_{i=1}^{k} \psi_i(v) \cdot c_i.$$

Putting together the above inequality and the right-hand-side of Eq. (14), it follows that:

$$C_v \geq p \cdot \sum_{i=0}^{k-1} \psi_i(v) - \sum_{i=0}^{k-1} \int_{\eta=L}^{v} \psi_i(\eta) \cdot d\eta -$$

$$\sum_{i=0}^{k-1} \psi_i(v) \cdot c_{i+1} - v \cdot \epsilon$$

$$\geq \text{ALG}\left(\mathcal{I}_v^{(\epsilon)}\right) - v \cdot \epsilon,$$

where ALG is the online algorithm corresponding to the set of allcation functions $\{\psi_i\}_{\forall i \in [k]}$ and recall that $\text{ALG}(\mathcal{I}_v^{(\epsilon)})$ is defined as follows:

$$\text{ALG}\left(\mathcal{I}_L^{(\epsilon)}\right) = \sum_{i=1}^{k} \psi_i(L) \cdot (L - c_i),$$

$$\text{ALG}\left(\mathcal{I}_v^{(\epsilon)}\right) = \text{ALG}\left(\mathcal{I}_L^{(\epsilon)}\right) + \sum_{i=1}^{k} \int_{\eta=L}^{v} (\eta - c_i) d\psi_i(\eta), \ \ \forall v \in [L, U].$$

Since $\{\psi_i\}_{\forall i \in [k]}$ satisfy Eq. (5), it follows that

$$C_v \geq \text{ALG}\left(\mathcal{I}_v^{(\epsilon)}\right) - v \cdot \epsilon$$

$$\geq \frac{1}{\alpha} \cdot \left(k \cdot v - \sum_{i=1}^{k} c_i\right) - v \cdot \epsilon, \quad \forall v \in [L, U].$$

By setting $\epsilon \to 0$, it follows that

$$C_v \geq \frac{1}{\alpha} \cdot \left(k \cdot v - \sum_{i=1}^{k} c_i\right), \quad \forall v \in [L, U].$$

To complete the proof of Lemma 2, we also need to prove that the above inequality holds as an equality for the set of functions $\{\psi_i^\alpha\}_{\forall i \in [k]}$. This can be proved by contradiction. Suppose that at some point $v \in [L, U]$, the above equality does not hold, then there must exist another set of feasible functions, say $\{\hat{\psi}_i\}_{\forall i \in [k]}$, induced by a new algorithm, say $\widehat{\text{ALG}}$, that satisfy Eq. (5) and

$$\sum_{i=1}^{k} \hat{\psi}_i(v) < \sum_{i=1}^{k} \psi_i^\alpha(v).$$

We argue that the new set of functions $\{\hat{\psi}_i\}_{\forall i \in [k]}$ will allocate a smaller fraction of its total units to buyers in $\mathcal{I}^{(\epsilon)}$ arriving at or before stage-$v$ compared to $\{\psi_i^\alpha\}_{\forall i \in [k]}$. However, by still following the allocation functions $\{\psi_i^\alpha\}_{\forall i \in [k]}$, $\widehat{\text{ALG}}(\mathcal{I}_v^{(\epsilon)})$ will be exactly equal to $\frac{1}{\alpha}(k \cdot v - \sum_{i=1}^{k} c_i)$. Given the definition of $\{\psi_i^\alpha\}_{\forall i \in [k]}$, we have $\sum_{i=1}^{k} \psi_i^\alpha(v) = \chi^\alpha(v)$, meaning that $\sum_{i=1}^{k} \hat{\psi}_i(v) < \chi^\alpha(v)$. However, this contradicts the definition of $\chi^\alpha(v)$. We thus complete the proof of Lemma 2.

## D  Proof of Proposition 2

From Lemma 2, we know that $\{\psi_i^\alpha(v)\}_{\forall i \in [k]}$ satisfy Eq. (5) with an equality. Therefore, the set of allocation functions $\{\psi_i^\alpha(v)\}_{\forall i \in [k]}$ is a solution to the following system of equations:

$$\sum_{i=1}^{k} \psi_i^\alpha(L) \cdot (L - c_i) + \sum_{i=1}^{k} \int_{\eta=L}^{v} (\eta - c_i) d\psi_i^\alpha(\eta)$$

$$= \frac{1}{\alpha} \cdot (k \cdot v - \sum_i c_i), \quad \forall i \in [k], v \in [L, U]. \tag{15}$$

Also, based on Lemma 3, we argue that if the value of the function $\psi_i^\alpha(v)$ is changing at some value $v \in [L, U]$ (i.e., $d\psi_i^\alpha(v) \neq 0$), then the value of all the functions $\{\psi_j^*(v)\}_{\forall j \in [i-1]}$ are equal to one, and all the functions in the set $\{\psi_j^*(v)\}_{j>i}$ are equal to zero. Based on this property, we can assign an interval $[\ell_i, u_i]$ to each $\psi_i^\alpha(v)$. In the interval of $[\ell_i, u_i]$, only the value of $\psi_i^\alpha$ changes while the other functions $\{\psi_j^\alpha\}_{\forall j \neq i}$ in that interval are fixed to be one or zero. Additionally, the following relation exists between the start and end points of these intervals:

$$L = \ell_1 \leq u_1 = \ell_2 \leq u_2 \leq \cdots \leq \ell_k \leq u_k = U.$$

To satisfy the equality $\sum_{i \in [k]} \psi_i^\alpha(L) \cdot (L - c_i) = \frac{1}{\alpha} \cdot (k \cdot L - \sum_i c_i)$, the set of functions $\{\psi_i^\alpha(v)\}_{\forall i \in [k-1]}$ should be equal to one at the point $v = L$. Thus, the explicit design of the functions $\{\psi_i^\alpha\}_{\forall i \in [k-1]}$ is as follows:

$$\psi_i^{(\alpha)}(v) = 1, \quad i = 1, \ldots, k-1.$$

In the case that $\sum_{i \in [k]} L - c_i < \frac{1}{\alpha} \cdot (k \cdot L - \sum_i c_i)$, to satisfy $\sum_{i \in [k]} \psi_i^\alpha(L) \cdot (L - c_i) = \frac{1}{\alpha} \cdot (k \cdot L - \sum_i c_i)$, we need to have:

$$\psi_{\underline{k}}^\alpha(L) = \frac{\sum_{i \in [\underline{k}-1]} (L - c_i) - \frac{1}{\alpha} \cdot \sum_{i \in [k]} (L - c_i)}{L - c_{\underline{k}}} = \xi.$$

Since for all $v \in [\ell_{\underline{k}}, u_{\underline{k}}]$ with $\ell_{\underline{k}} = L$, only the value of $\psi_{\underline{k}}^\alpha(v)$ changes (i.e., $d\psi_i^\alpha(v) = 0$ for all $i \neq \underline{k}$), it follows that:

$$\sum_{i=1}^{k} \psi_i^\alpha(L) \cdot (L - c_i) + \sum_{i=1}^{k} \int_{\eta=L}^{v} (\eta - c_i) d\psi_i^\alpha(\eta)$$

$$= \sum_{i=1}^{k} \psi_i^\alpha(L) \cdot (L - c_i) + \int_{\eta=L}^{v} (\eta - c_{\underline{k}}) d\psi_{\underline{k}}^*(\eta), \ \forall v \in [L, u_{\underline{k}}].$$

Based on the system of equations in Eq. (15), we need to have:

$$\sum_{i=1}^{k} \psi_i^\alpha(L) \cdot (L - c_i) + \int_{\eta=L}^{v} (\eta - c_{\underline{k}}) d\psi_{\underline{k}}^*(\eta)$$

$$= \frac{1}{\alpha} \cdot (k \cdot v - \sum_i c_i), \qquad \forall v \in [\ell_k, u_{\underline{k}}].$$

Taking derivative w.r.t. $v$ from both sides of the equation above, we have

$$(v - c_{\underline{k}}) \cdot d\psi_{\underline{k}}^*(v) = \frac{k}{\alpha}.$$

Solving the above differential equation leads to

$$\psi_{\underline{k}}^*(v) = \frac{k}{\alpha} \cdot \ln(v - c_{\underline{k}}) + Q, \quad \forall v \in [\ell_{\underline{k}}, u_{\underline{k}}],$$

where $Q$ is a constant. To find $Q$, since $\psi_{\underline{k}}^*(L) = \xi$, it follows that $Q = \xi - \frac{k}{\alpha} \cdot \ln(L - c_{\underline{k}})$. As a result, the explicit design of the function $\psi_{\underline{k}}^\alpha$ is as follows:

$$\psi_{\underline{k}}^{(\alpha)}(v) = \begin{cases} \xi + \frac{k}{\alpha} \cdot \ln\left(\frac{v - c_{\underline{k}}}{L - c_{\underline{k}}}\right) & v \in [L, u_{\underline{k}}], \\ 1 & v > u_{\underline{k}}. \end{cases}$$

To obtain the value of $u_{\underline{k}}$, we set $\psi_{\underline{k}}^*(u_{\underline{k}}) = 1$ (the function $\psi_{\underline{k}}^*$ reaches its maximum). Consequently, it follows that:

$$u_{\underline{k}} = (L - c_{\underline{k}}) \cdot e^{\frac{\alpha}{k} \cdot (1 - \xi)} + c_{\underline{k}}.$$

Using the same procedure as what has been applied to $\psi_{\underline{k}}^\alpha$, for all the other functions $\{\psi_i^\alpha(v)\}_{\forall i > \underline{k}}$, we have

$$\sum_{j=1}^{k} \psi_j^*(L) \cdot (L - c_j) + \sum_{j=1}^{k} \int_{\eta=L}^{v} (\eta - c_j) d\psi_i^\alpha(\eta)$$

$$= \sum_{j=1}^{k} \psi_j^*(L) \cdot (L - c_j) + \int_{\eta=L}^{v} (\eta - c_i) d\psi_i^\alpha(\eta), \quad \forall v \in [\ell_i, u_i].$$

Taking derivative w.r.t. $v$ from both sides of the equation above, it follows that:

$$(v - c_i) \cdot d\psi_i^\alpha(v) = \frac{k}{\alpha}.$$

Solving the above differential equation leads to

$$\psi_i^\alpha(v) = \frac{k}{\alpha} \cdot \ln(v - c_i) + \hat{Q}, \quad \forall v \in [\ell_i, u_i].$$

Since $\psi^*(\ell_i) = 0$, we have $\hat{Q} = -\ln(\ell_i - c_i)$. The explicit design of the function $\psi_i^\alpha$ is thus as follows:

$$\psi_i^{(\alpha)}(v) = \begin{cases} 0 & v \leq \ell_i, \\ \frac{k}{\alpha} \cdot \ln\left(\frac{v - c_i}{\ell_i - c_i}\right) & v \in [\ell_i, u_i], \quad i = \underline{k} + 1, \ldots, k - 1 \\ 1 & v \geq u_i. \end{cases}$$

For the function $\psi_k^\alpha$, since it is the last function, it follows that:

$$\psi_k^{(\alpha)}(v) = \begin{cases} 0 & v \leq \ell_k, \\ \frac{k}{\alpha} \cdot \ln\left(\frac{v - c_k}{\ell_k - c_k}\right) & v \in [\ell_k, U]. \end{cases}$$

By setting $\psi^*(u_i) = 1$, it follows that:

$$u_i = (\ell_i - c_i) \cdot e^{\frac{\alpha}{k}} + c_i, \quad \underline{k} + 1 \leq i \leq k.$$

Putting everything together, Proposition 2 follows.

# E Full Proof of Theorem 5

In this section, we provide a complete proof of Theorem 5. We begin by introducing several important notations and lemmas. Then, we break the problem into two independent subproblems based on the buyers' valuations in some arbitrary arrival instance $\mathcal{I}$. For each case, we proceed to show how to upper bound $\text{OPT}(\mathcal{I})$, the objective of the optimal offline algorithm on $\mathcal{I}$. We then proceed to lower bound the expected performance of r-Dynamic on that instance, $\text{ALG}(\mathcal{I})$. Ultimately, we combine everything and obtain a performance guarantee for r-Dynamic under all adversarially chosen instances of OSDoS for that subproblem.

## E.1 Notations and Definitions

Consider an arbitrary arrival instance $\mathcal{I} = \{v_t\}_{t \in [T]}$. Recall that the random price vector $\mathbf{P} = \{P_1, \cdots, P_k\}$ is generated using the pricing functions $\{\phi_i\}_{\forall i \in [k]}$ at the beginning of r-Dynamic (line 3 of Algorithm 1). In the following, we will refer to Algorithm 1 as r-Dynamic($\mathbf{P}$) to indicate that the algorithm is executed with the random price vector being realized as $\mathbf{P}$. Based on the design of $\{\phi_i\}_{\forall i \in [k]}$ in Theorem 5, the first $k^* - 1$ prices in $\mathbf{P}$ are all $L$'s (i.e., $P_1 = \cdots = P_{k^*-1} = L$), the $k^*$-th price $P_{k^*}$ is a random variable within $[L, U_{k^*}]$, and for all $i \in \{k^* + 1, \cdots, k\}$, we have $P_i \in [L_i, U_i]$ (recall that $P_i$ is also a random variable). Here, the values of $k^*$ and $\{[L_i, U_i]\}_{\forall i}$ are all defined in Theorem 5.

Let $\mathcal{P}$ denote the support of all possible values of the random price vector $\mathbf{P}$:

$$\mathcal{P} = \{L\}^{k^*-1} \times [L, U_{\underline{k}^*}] \times \prod_{i \in \{\underline{k}^*+1, \cdots, k\}} [L_i, U_i].$$

Given a price vector realization $\mathbf{P} \in \mathcal{P}$, let $W(\mathbf{P})$ represent the total number of items allocated by r-Dynamic($\mathbf{P}$) under the input instance $\mathcal{I}$. Since $\mathbf{P}$ is a random variable, $W(\mathbf{P})$ is also a random variable. For clarity, we will sometimes omit the price vector and refer to it simply as $W$ whenever the context is clear.

Let $\omega$ denote the maximum value in the support of the random variable $W$ (i.e., $\omega$ is the maximum possible value of $W(\mathbf{P})$ for all $\mathbf{P} \in \mathcal{P}$). Thus, $\omega$ is a deterministic value that depends only on the input instance $\mathcal{I}$. Furthermore, let $\boldsymbol{\pi} \in \mathcal{P}$ be a price vector such that r-Dynamic($\boldsymbol{\pi}$) allocates the $\omega$-th item earlier than any other price vector in the set $\mathcal{P}$. That is, for all $\mathbf{P} \in \mathcal{P}$, r-Dynamic($\mathbf{P}$) allocates the $\omega$-th item no earlier than that of r-Dynamic($\boldsymbol{\pi}$).

Let us define the set $\{(v_i, \tau_i)\}_{\forall i \in [\omega]}$ so that $\tau_i$ is the arrival time of the buyer in the instance $\mathcal{I}$ to whom r-Dynamic($\boldsymbol{\pi}$) allocates the $i$-th unit and $v_i$ is its valuation. Note that for all $i \in \{1, \cdots, \omega\}$, $\tau_i$ and $v_i$ are deterministic values once $\boldsymbol{\pi}$ and $\mathcal{I}$ are given.

We can derive the following inequality regarding $v_i$:

$$v_i \geq L_i, \quad \forall i \in [\omega], \tag{16}$$

where $L_i$ is the lower bound for the range of the pricing function $\phi_i$, used to generate the random price for the $i$-th unit. This inequality holds since the buyer arriving at time $\tau_i$ accepts the price posted for the $i$-th unit by r-Dynamic. The price for the $i$-th unit is at least equal to $L_i$ based on the design of the pricing functions $\phi_i$.

Let the random variable $W^{\tau_\omega}(\mathbf{P})$ denote the total number of items allocated by r-Dynamic($\mathbf{P}$) after the arrival of buyer $\tau_\omega$ in the instance $\mathcal{I}$. The lemma below shows that the random variable $W^{\tau_\omega}(\mathbf{P})$ is always lower bounded by $\omega - 1$.

LEMMA 7. Given an arbitrary instance $\mathcal{I}$, $W^{\tau_\omega}(\mathbf{P}) \geq \omega - 1$ holds for all $\mathbf{P} \in \mathcal{P}$.

PROOF. If $\omega = 1$, this lemma is trivial, so we consider the case where $\omega \geq 2$. Suppose before the arrival of the buyer at time $\tau_2$, no items have been sold. From Eq. (16), we know that $v_2 \geq L_2$. Additionally, based on the design of the pricing functions $\phi_1(.)$ and $\phi_2(.)$, we have $L_2 \geq U_1$. Consequently, it follows that $v_2 \geq U_1$. Since the realized price for the first unit under any sampled price vector will be at most $U_1$ (based on design of the pricing function $\phi_1$), the buyer arriving at time $\tau_2$ will accept the price for the first unit, and the algorithm will sell the first item. Thus, for all possible

price vector $\mathbf{P}$, the value of the random variable $W^{\tau_2}(\mathbf{P})$ is at least equal to one. By the same reasoning, if before the arrival of the buyer at time $\tau_3$, only one item has been sold, the buyer arriving at $\tau_3$ will accept the price for the second unit, regardless of its price, and the total number of items sold by r-Dynamic will increase to two. This reasoning can be extended to the time $\tau_\omega$. As a result, after the arrival of the buyer at time $\tau_\omega$, r-Dynamic sells at least $\omega - 1$ units and thereby the claim in the lemma follows. $\quad\square$

Lemma 7 implies that the support of the random variable $W^{\tau_\omega}$ consists only of two values: $\omega - 1$ and $\omega$. This greatly simplifies the analysis of the algorithm.

The following two lemmas help us lower bound the expected performance of r-Dynamic under the input instance $\mathcal{I}$ and upper bound the objective of the offline optimal algorithm given the instance $\mathcal{I}$, respectively.

LEMMA 8. *If a buyer in instance $\mathcal{I}$ arrives before time $\tau_\omega$ with a valuation within $[L_\omega, U]$, then for all $\mathbf{P} \in \mathcal{P}$, r-Dynamic($\mathbf{P}$) will allocate one unit of the item to that buyer.*

PROOF. According to the definition of $\pi$, $\tau_\omega$ is the earliest time across all possible price vectors in $\mathcal{P}$ that the production level exceeds $\omega - 1$, causing the posted price to exceed $U_{\omega-1}$. Thus, for all possible realization of $\mathbf{P}$, the posted prices by r-Dynamic remain below $U_{\omega-1}$ before the arrival of buyer at time $\tau_\omega$. Consequently, when a buyer with a valuation within $[L_\omega, U]$ arrives before time $\tau_\omega$, the buyer accepts the price posted to him (since $L_\omega \geq U_{\omega-1}$) and a unit of item will thus be allocated to this buyer. $\quad\square$

LEMMA 9. *There are no buyers in instance $\mathcal{I}$ with a valuation within $[U_\omega, U]$ arriving after time $\tau_\omega$, namely, the valuations of all buyers arrive after $\tau_\omega$ are less than $U_\omega$.*

PROOF. If there exist a buyer with a valuation larger than $U_\omega$ arriving after the time $\tau_\omega$, then there must exist a price vector in $\mathcal{P}$, say $\mathbf{P}'$, such that the number of units sold by r-Dynamic($\mathbf{P}'$) will exceed $\omega$. This contradicts the definition of $\omega$. Thus, the lemma follows.[2] $\quad\square$

Given an instance $\mathcal{I}$, let the set $\mathcal{B} \subseteq \mathcal{I}$ contain the highest-valued buyers that the offline optimal algorithm selects. We further divide $\mathcal{B}$ into two subsets: $\mathcal{B}_1$ and $\mathcal{B}_2$. $\mathcal{B}_1$ comprises the highest-valued buyers up to time $\tau_\omega$, while $\mathcal{B}_2$ includes the remaining buyers in $\mathcal{B}$ who arrive at or after time $\tau_\omega$. Let us further partition $\mathcal{B}_1$ into two subsets: $\mathcal{B}_{1,1}$ and $\mathcal{B}_{1,2}$. Here, $\mathcal{B}_{1,1}$ consists of buyers in $\mathcal{B}_1$ with valuations at least $L_\omega$, and $\mathcal{B}_{1,2} = \mathcal{B}_1 \setminus \mathcal{B}_{1,1}$ comprises those with valuations strictly less than $L_\omega$.

For the rest of the analysis, let us study the problem for two separate cases that may occur depending on the instance $\mathcal{I}$.

## E.2 Case 1: Buyer $\tau_\omega$ Has the Highest Valuation

In this case, in the set $\mathcal{B}_2$, no buyer has a valuation greater than $U_{\omega-1}$ except for the buyer at time $\tau_\omega$. Therefore, the buyer at time $\tau_\omega$ possesses the highest valuation in the instance $\mathcal{I}$.

---

[2]In fact, such a price vector $\mathbf{P}'$ for the initial $\omega$ units should have the same prices as the vector $\pi$ and for the $(i + 1)$-th unit, $\mathbf{P}'$ should be equal to $U^\omega$ (i.e., $P'_{i+1} = U^\omega$).

*E.2.1 Bound OPT from Above for Case 1.* The following upper bound can be derived for $\text{OPT}(\mathcal{I})$, which denotes the objective value of the offline optimal algorithm on instance $\mathcal{I}$:

$$\text{OPT}(\mathcal{I})$$
$$= V(\mathcal{B}_1) + V(\mathcal{B}_2) - \sum_{i=1}^{|\mathcal{B}|} c_i$$
$$\leq V(\mathcal{B}_1) + (|\mathcal{B}_2| - 1) \cdot U_{\omega-1} + v_{\tau_\omega} - \sum_{i=1}^{|\mathcal{B}|} c_i$$
$$= V(\mathcal{B}_{1,1}) + V(\mathcal{B}_{1,2}) + (|\mathcal{B}_2| - 1) \cdot U_{\omega-1} + v_{\tau_\omega} - \sum_{i=1}^{|\mathcal{B}|} c_i$$
$$\leq |\mathcal{B}_{1,1}| \cdot U_{\omega-1} + (V(\mathcal{B}_{1,1}) - |\mathcal{B}_{1,1}| \cdot U_{\omega-1})$$
$$\qquad + |\mathcal{B}_{1,2}| \cdot U_{\omega-1} + (|\mathcal{B}_2| - 1) \cdot U_{\omega-1}$$
$$\qquad + v_{\tau_\omega} - \sum_{i=1}^{|\mathcal{B}|} c_i$$
$$\leq (k - 1) \cdot U_{\omega-1} + (V(\mathcal{B}_{1,1}) - |\mathcal{B}_{1,1}| \cdot U_{\omega-1})$$
$$\qquad + v_{\tau_\omega} - \sum_{i=1}^{k} c_i,$$

where the first inequality directly follows the condition of **Case 1**. The second inequality follows the definition of $\mathcal{B}_{1,1}$ and $\mathcal{B}_{1,2}$. Finally, the third inequality follows the fact that we only focus on the case when $c_k < L$.

*E.2.2 Bound ALG from Below for Case 1.* Moving forward, we focus on establishing a lower bound on the performance of r-Dynamic under the arrival instance $\mathcal{I}$. Let the random variables $\{X_i\}_{\forall i \in [k]}$ represent the value obtained by r-Dynamic from allocating the $i$-th unit of the item. Given the input instance $\mathcal{I}$, let $\mathbb{E}[\text{ALG}(\mathcal{I})]$ denote the expected performance of r-Dynamic. Therefore, we have:

$$\mathbb{E}[\text{ALG}(\mathcal{I})]$$
$$= \mathbb{E}\left[ \sum_{i=1}^{k} (X_i - c_i) \cdot \mathbb{1}_{\{i\text{-th item is sold under price vector } \mathbf{P}\}} \right],$$
$$\geq \sum_{i=1}^{\omega-1} \mathbb{E}[X_i - c_i]$$
$$= \sum_{i=1}^{\omega-1} \mathbb{E}[X_i] - \sum_{i=1}^{\omega-1} c_i$$
$$\geq \sum_{i=1}^{\omega-1} \int_0^1 \phi_i(\eta) d\eta + (V(\mathcal{B}_{1,1}) - |\mathcal{B}_{1,1}| \cdot U_{\omega-1}) - \sum_{i=1}^{\omega-1} c_i.$$

In the equations above, all expectations are taken with respect to the randomness of the price vector $\mathbf{P}$. The first inequality follows Lemma 7, indicating that under any price vector $\mathbf{P}$, r-Dynamic sells at least $\omega - 1$ units. The first term in the second inequality follows due to the independent sampling used to set the price of the $i$-th unit using the pricing function $\phi_i$, and the second term follows Lemma 8.

Let us define $\psi_i(v) = \sup\{s : \phi_i(s) \leq v\}$ for all $i \in [k]$. From the definition of $\{\phi_i\}_{\forall i \in [k]}$ in Theorem 5, it follows that:

$$\mathbb{E}[\text{ALG}(\mathcal{I})]$$

$$\geq \sum_{i=1}^{\omega-1} \int_0^1 \phi_i(\eta) d\eta \ - \ \sum_{i=1}^{\omega-1} c_i + \left(V(\mathcal{B}_{1,1}) - |\mathcal{B}_1| \cdot L_\omega\right)$$

$$= \sum_{i=1}^{\omega} \psi_i(L) \cdot (L - c_{i+1}) + \sum_{i=1}^{\omega-1} \int_{\eta=L}^{U_{\omega-1}} (\eta - c_{i+1}) d\psi_i(\eta)$$

$$+ \left(V(\mathcal{B}_{1,1}) - |\mathcal{B}_1| \cdot L_\omega\right).$$

Furthermore, it is evident that based on the design of $\{\phi_i\}_{\forall i \in [k]}$ with $\alpha = \alpha_{\mathcal{S}}^*(k)$, the set of functions $\{\psi_i(v)\}_{\forall i \in [k]}$ follows the same design as $\{\psi_i^\alpha(v)\}_{\forall i \in [k]}$ given in Proposition 2. As a result, it follows that:

$$\sum_{i=1}^{k} \psi_i(L) \cdot (L - c_{i+1}) + \sum_{i=1}^{\omega-1} \int_{\eta=L}^{U_{\omega-1}} (\eta - c_{i+1}) d\psi_i(\eta)$$

$$+ \left(V(\mathcal{B}_1) - |\mathcal{B}_1| \cdot L_\omega\right)$$

$$\geq \frac{1}{\alpha_{\mathcal{S}}^*(k)} \cdot \left(k \cdot U_{\omega-1} - \sum_i c_i\right) + \left(V(\mathcal{B}_{1,1}) - |\mathcal{B}_1| \cdot L_\omega\right).$$

*E.2.3 Putting Everything Together for Case 1.* Putting together the lower bound and upper bound derived for the expected objective value of r-Dynamic and the offline optimal algorithm, it follows that:

$$\frac{\text{OPT}(\mathcal{I})}{\mathbb{E}[\text{ALG}(\mathcal{I})]}$$

$$\leq \frac{(k-1) \cdot U_{\omega-1} + v_{\tau_\omega} + (V(\mathcal{B}_{1,1}) - |\mathcal{B}_{1,1}| \cdot L_\omega) - \sum_{i=1}^{k} c_i}{\frac{1}{\alpha_{\mathcal{S}}^*(k)} \cdot (k \cdot U_{\omega-1} - \sum_i c_i) + (V(\mathcal{B}_{1,1}) - |\mathcal{B}_{1,1}| \cdot L_\omega)}$$

$$\leq \frac{(k-1) \cdot U_{\omega-1} + v_{\tau_\omega} - \sum_{i=1}^{k} c_i}{\frac{1}{\alpha_{\mathcal{S}}^*(k)} \cdot (k \cdot U_{\omega-1} - \sum_i c_i)}$$

$$= \alpha_{\mathcal{S}}^*(k) \cdot \left(1 + \frac{v_{\tau_\omega} - U_{\omega-1}}{k \cdot U_{\omega-1} - C}\right)$$

$$\leq \alpha_{\mathcal{S}}^*(k) \cdot \left(1 + \frac{U_\omega - U_{\omega-1}}{k \cdot U_{\omega-1} - C}\right)$$

$$\leq \alpha_{\mathcal{S}}^*(k) \cdot e^{\frac{\alpha_{\mathcal{S}}^*(k)}{k}}.$$

In the equation above, the last inequality is due to the fact that $\frac{U_\omega - U_{\omega-1}}{U_{\omega-1} - c_\omega} = \frac{U_\omega - c_\omega}{U_{\omega-1} - c_\omega} - 1 \leq 1 + e^{\frac{\alpha_{\mathcal{S}}^*(k)}{k}}$, where the last inequality follows the design in Eq. (10).

## E.3 Case 2: Buyer $\tau_\omega$ Does Not Have the Highest Valuation

In the set of buyers $\mathcal{B}_2$, there are other buyers with valuation greater than $U_{\omega-1}$ besides the buyer at time $\tau_\omega$. Let $\lambda$ denote the value of the highest buyer in $\mathcal{B}_2$ along with the value of buyer at time $\tau_\omega$. First, let us consider the case that $\lambda \leq v_{\tau_\omega}$. The proof for the case that $\lambda > v_{\tau_\omega}$ follows exactly the same as the following case.

*E.3.1 Bound OPT from Above for Case 2.* Following the same approach as the previous Case 1, let us first upper bound the objective

of the offline optimal algorithm on instance $\mathcal{I}$:

$$\text{OPT}(\mathcal{I})$$

$$= V(\mathcal{B}_1) + V(\mathcal{B}_2) - \sum_{i=1}^{|\mathcal{B}|} c_i$$

$$\leq V(\mathcal{B}_1) + (|\mathcal{B}_2| - 1) \cdot \lambda + v_{\tau_\omega} - \sum_{i=1}^{|\mathcal{B}|} c_i$$

$$\leq V(\mathcal{B}_{1,1}) + V(\mathcal{B}_{1,2}) + (|\mathcal{B}_2| - 1) \cdot \lambda + v_{\tau_\omega} - \sum_{i=1}^{|\mathcal{B}|} c_i$$

$$\leq |\mathcal{B}_{1,1}| \cdot U_{\omega-1} + (V(\mathcal{B}_1) - |\mathcal{B}_{1,1}| \cdot U_{\omega-1}) +$$

$$|\mathcal{B}_{1,2}| \cdot U_{\omega-1} + (|\mathcal{B}_2| - 1) \cdot \lambda + v_{\tau_\omega} - \sum_{i=1}^{|\mathcal{B}|} c_i$$

$$\leq (k-1) \cdot \lambda + v_{\tau_\omega} + (V(\mathcal{B}_1) - |\mathcal{B}_1| \cdot L_\omega) - \sum_{i=1}^{k} c_i.$$

*E.3.2 Bound ALG from Below for Case 2.* To establish a lower bound on the performance of r-Dynamic in this case, let us consider the following lemma:

LEMMA 10. *If the random price of the $\omega$-th unit is realized to be less than $\lambda$ and further assume that $\lambda \leq v_{\tau_\omega}$, then the number of items allocated by r-Dynamic in the end is equal to $\omega$.*

PROOF. Under any price realization, as established by Lemma 7, it is proven that after the arrival of the buyer at time $\tau_\omega$, the number of allocated units is at least $\omega - 1$. If the price of the $\omega$-th unit is realized to be less than $\lambda$, then upon the arrival of the buyer with valuation $\lambda$ at some time after $\tau_\omega$, the buyer will accept the price if the $\omega$-th unit has not already been sold. □

Next, we obtain a lower bound on the performance of r-Dynamic as follows:

$$\mathbb{E}[\text{ALG}(\mathcal{I})]$$

$$= \mathbb{E}\left[\sum_{i=1}^{k} (X_i - c_i) \cdot \mathbb{1}_{\{i\text{-th item is sold under pricie vector P}\}}\right]$$

$$\geq \sum_{i=1}^{\omega-1} \mathbb{E}[X_i - c_i] + \mathbb{E}[X_\omega - c_\omega | P_\omega \leq \lambda]$$

$$\geq \sum_{i=1}^{\omega-1} \int_0^1 \phi_i(\eta) d\eta \ + \int_0^{\phi_\omega^{-1}(\lambda)} \phi_\omega(\eta) d\eta - \phi_\omega^{-1}(\lambda) \cdot c_\omega$$

$$- \sum_{i=1}^{\omega-1} c_i + (V(\mathcal{B}_1) - |\mathcal{B}_1| \cdot L_\omega).$$

In the equations above, all expectations are taken with respect to the randomness of the price vector $\mathbf{P} \in \mathcal{P}$. The first inequality follows Lemma 10, where $P_\omega$ denotes the $\omega$-element of the random price vector $\mathbf{P}$ that r-Dynamic posts for the $\omega$-th unit. The second inequality is true because of the independent sampling that is used to set the random price of the $i$-th unit using $\phi_i$ and Lemma 8.

Let us define $\psi_i(v) = \sup\{s : \phi_i(s) \leq v\}$, $i \in [k]$. From the definition of $\{\phi_i\}_{i \in [k]}$ in Theorem 5, it follows that:

$$\sum_{i=1}^{\omega-1} \int_0^1 \phi_i(\eta) d\eta + \int_0^{\phi_\omega^{-1}(\lambda)} \phi_\omega(\eta) d\eta - \phi_\omega^{-1}(\lambda) \cdot c_\omega$$

$$- \sum_{i=1}^{\omega-1} c_i + (V(\mathcal{B}_1) - |\mathcal{B}_1| \cdot L_\omega)$$

$$= \sum_{i=1}^{k} \psi_i(L) \cdot (L - c_{i+1}) + \sum_{i=1}^{\omega} \int_{\eta=L}^{\lambda} (\eta - c_{i+1}) d\psi_i(\eta)$$

$$+ (V(\mathcal{B}_1) - |\mathcal{B}_1| \cdot L_\omega).$$

Furthermore, it is evident that the set of functions $\{\psi_i(v)\}_{\forall i \in [k]}$ follows the same design as $\{\psi_i^\alpha(v)\}_{i \in [k]}$ given in Lemma 2 (recall that $\{\psi_i^\alpha(v)\}_{i \in [k]}$ are based on $\{\phi_i\}_{\forall i \in [k]}$). As a result, it follows that:

$$\sum_{i=1}^{k} \psi_i(L) \cdot (L - c_{i+1}) + \sum_{i=1}^{\omega} \int_{\eta=L}^{\lambda} (\eta - c_{i+1}) d\psi_i(\eta)$$

$$+ (V(\mathcal{B}_1) - |\mathcal{B}_1| \cdot L_\omega)$$

$$\geq \frac{1}{\alpha_{\mathcal{S}}^*(k)} \cdot \left( k \cdot \lambda - \sum_i c_i \right) + (V(\mathcal{B}_1)) - |\mathcal{B}_1| \cdot L_\omega).$$

*E.3.3 Putting Everything Together for Case 2.* Putting together the above lower and upper bounds, it follows that:

$$\frac{\text{OPT}(\mathcal{I})}{\mathbb{E}[\text{ALG}(\mathcal{I})]}$$

$$\leq \frac{(k-1) \cdot \lambda + v_{\tau_\omega} + (V(\mathcal{B}_1) - |\mathcal{B}_1| \cdot L_\omega) - \sum_{i=1}^{k} c_i}{\frac{1}{\alpha_{\mathcal{S}}^*(k)} \cdot (k \cdot \lambda - \sum_{i=1}^{k} c_i) + (V(\mathcal{B}_1) - |\mathcal{B}_1| \cdot L_\omega)}$$

$$\leq \frac{(k-1) \cdot \lambda + v_{\tau_\omega} - \sum_{i=1}^{k} c_i}{\frac{1}{\alpha_{\mathcal{S}}^*(k)} \cdot (k \cdot \lambda - \sum_{i=1}^{k} c_i)}$$

$$= \alpha_{\mathcal{S}}^*(k) \cdot (1 + \frac{v_{\tau_\omega} - \lambda}{k \cdot \lambda - C})$$

$$\leq \alpha_{\mathcal{S}}^*(k) \cdot (1 + \frac{v_{\tau_\omega} - \lambda}{k \cdot \lambda - C})$$

$$\leq \alpha_{\mathcal{S}}^*(k) \cdot (1 + \frac{U_\omega - U_{\omega-1}}{k \cdot U_{\omega-1} - C})$$

$$\leq \alpha_{\mathcal{S}}^*(k) \cdot e^{\frac{\alpha_{\mathcal{S}}^*(k)}{k}}.$$

We thus complete the proof of Theorem 5.

REMARK 1. *Theorem 5 argues that r-Dynamic is asymptotically optimal. We emphasize that our analysis of Theorem 5 is not tight because it does not differentiate between the sample paths of r-Dynamic when the algorithm sells $\omega - 1$ units and those when it sells $\omega$ units. As a result, our analysis considers that r-Dynamic sells $\omega - 1$ units of the item on all sample paths.[3] However, in the subsequent analysis for the case of $k = 2$, we can enumerate all the scenarios and therefore do not require such a reduction. For this reason, we can prove in the next section that r-Dynamic is indeed optimal for $k = 2$ (see the proof of Corollary 6 next).*

---

[3]We conjecture that r-Dynamic is optimal even in the small inventory regime if a tighter analysis is performed.

# F  Proof of Corollary 6

In this section, we prove that for an arbitrary instance $\mathcal{I}$, the expected performance of r-Dynamic, denoted as $\mathbb{E}[\text{ALG}(\mathcal{I})]$, is at least $\frac{\text{OPT}(\mathcal{I})}{\alpha_{\mathcal{S}}^*(2)}$.

Let $v_1^*, v_2^*$ denote the two highest valuations in the instance $\mathcal{I}$ (we omit the proof for the trivial case with only one buyer in $\mathcal{I}$). Depending on the values of $v_1^*$ and $v_2^*$, the following three cases occur. In each scenario, we prove that $\mathbb{E}[\text{ALG}(\mathcal{I})] \geq \frac{\text{OPT}(\mathcal{I})}{\alpha_{\mathcal{S}}^*(2)} = \frac{v_1^* + v_2^* - c_1 - c_2}{\alpha_{\mathcal{S}}^*(2)}$ holds.

**Case I:** $v_1^* \leq v_2^* \leq U_1$. Let random variables $X_1$ and $X_2$ denote the valuations of the buyers that purchase the first and second unit of the item, respectively. Then, it follows that

$$\mathbb{E}[\text{ALG}(\mathcal{I})]$$

$$= \mathbb{E}_{s \sim U^2(0,1)}[X_1 + X_2 - c_1 - c_2]$$

$$\geq \int_{s_1=0}^{\phi_1^{-1}(v_1^*)} (\phi_1(s_1) - c_1) \cdot ds_1$$

$$+ (v_2^* - c_1) \cdot (\phi_1^{-1}(v_2^*) - \phi_1^{-1}(v_1^*)),$$

$$\geq \int_{s_1=0}^{\phi_1^{-1}(v_1^*)} (\phi_1(s_1) - c_1) \cdot ds_1$$

$$+ \int_{s_1=\phi_1^{-1}(v_1^*)}^{\phi_1^{-1}(v_2^*)} (\phi_1(s_1) - c_1) \cdot ds_1$$

$$= \int_{s_1=0}^{\phi_1^{-1}(v_2^*)} (\phi_1(s_1) - c_1) \cdot ds_1$$

$$\geq \frac{v_1^* + v_2^* - c_1 - c_2}{\alpha_{\mathcal{S}}^*(2)}$$

$$= \frac{\text{OPT}(\mathcal{I})}{\alpha_{\mathcal{S}}^*(2)},$$

where the first two terms in the first inequality arise from the fact that if the realized price for the first unit of the item, denoted as $P_1 = \phi_1(s_1)$, is set below $v_1^*$, then in the worst-case scenario, the value obtained from the first item will be at least equal to $\phi_1(s_1)$. The subsequent two terms are included because if the price for the first item falls within the range from $v_1^*$ to $v_2^*$, then the first item is allocated to the buyer whose valuation is $v_2^*$. The second inequality follows since $\phi_1(s_1)$ is an non-decreasing function. The third inequality follows from the design of $\phi_1(s_1)$ in Theorem 6.

**Case II:** $v_1^* \leq U_1 = L_2 \leq v_2^* \leq U$. In this case, we have

$$\mathbb{E}[\text{ALG}(\mathcal{I})]$$

$$= \mathbb{E}_{s \sim U^2(0,1)}[X_1 - c_1] + \mathbb{E}_{s \sim U^2(0,1)}[X_2 - c_2]$$

$$\geq \int_{s_1=0}^{\phi_1^{-1}(v_1^*)} (\phi_1(s_1) - c_1) ds_1 + (v_2^* - c_1) \cdot (1 - \phi_1^{-1}(v_1^*))$$

$$+ (v_2^* - c_2) \cdot \phi_1^{-1}(v_1^*) \cdot \phi_2^{-1}(v_2^*)$$

$$= \frac{2 \cdot v_1^* - c_1 - c_2}{\alpha_{\mathcal{S}}^*(2)} + (v_2^* - c_1) \cdot (1 - \phi_1^{-1}(v_1^*))$$

$$+ (v_2^* - c_2) \cdot \phi_1^{-1}(v_1^*) \cdot \phi_2^{-1}(v_2^*).$$

To prove $\mathbb{E}[\text{ALG}(I)] \geq \frac{\text{OPT}(I)}{\alpha_S^*(2)} = \frac{v_1^* + v_2^* - c_1 - c_2}{\alpha_S^*(2)}$, we define the following function

$$G(v_1^*, v_2^*) = \frac{2 \cdot v_1^* - c_1 - c_2}{\alpha_S^*(2)} + (v_2^* - c_1) \cdot (1 - \phi_1^{-1}(v_1^*)) +$$

$$(v_2^* - c_2) \cdot \phi_1^{-1}(v_1^*) \cdot \phi_2^{-1}(v_2^*) - \frac{v_1^* + v_2^* - c_1 - c_2}{\alpha_S^*(2)}.$$

Then the goal is to prove $G(v_1^*, v_2^*) \geq 0$ in its domain $L \leq v_1^* \leq U_1$ and $L_2 \leq v_2^* \leq U$. The proposition below formally states this result.

PROPOSITION 3. *For all $v_1^* \in [L_1, U_1]$ and $v_2^* \in [L_2, U_2]$, we have $G(v_1^*, v_2^*) \geq 0$.*

We deferred the proof of the above proposition to Appendix G. The idea is to simply prove that $G(v_1^*, v_2^*) \geq 0$ holds at all extreme points within its domain.

**Case III:** $L_2 \leq v_1^* \leq v_2^*$. In this case, we show that we can lower bound the expected performance of r-Dynamic as follows:

$$\mathbb{E}[\text{ALG}(I)]$$

$$= \mathbb{E}_{s \sim U^2(0,1)}[X_1 - c_1] + \mathbb{E}_{s \sim U^2(0,1)}[X_2 - c_2]$$

$$\geq \int_{s_1=0}^{1} (\phi_1(s_1) - c_1) \cdot ds_1 + \int_{s_2=0}^{\phi_2^{-1}(v_1^*)} (\phi_2(s_2) - c_2) \cdot ds_2$$

$$+ (v_2^* - c_2) \cdot (\phi_2^{-1}(v_2^*) - \phi_2^{-1}(v_1^*))$$

$$\geq \int_{s_1=0}^{1} (\phi_1(s_1) - c_1) \cdot ds_1 + \int_{s_2=0}^{\phi_2^{-1}(v_1^*)} (\phi_2(s_2) - c_2) \cdot ds_2$$

$$+ \int_{s_2=\phi_2^{-1}(v_1^*)}^{\phi_2^{-1}(v_2^*)} (\phi_2(s_2) - c_2) \cdot ds_2$$

$$= \int_{s_1=0}^{1} (\phi_1(s_1) - c_1) \cdot ds_1 + \int_{s_2=0}^{\phi_2^{-1}(v_2^*)} (\phi_2(s_2) - c_2) \cdot ds_2$$

$$= \frac{2 \cdot U_1 - c_1 - c_2}{\alpha_S^*(2)} + \int_{s_2=0}^{\phi_2^{-1}(v_2^*)} (\phi_2(s_2) - c_2) \cdot ds_2$$

$$= \frac{2 \cdot v_2^* - c_1 - c_2}{\alpha_S^*(2)}$$

$$\geq \frac{\text{OPT}(I)}{\alpha_S^*(2)},$$

where the first term in the first inequality arises from the fact that if the realized price for the first unit of the item, denoted as $P_1 = \phi_1(s_1)$, is set below $L_2$, then in the worst-case scenario, the value obtained from the first item will be at least equal to $\phi_1(s_1)$. The second and third terms follow the same reasoning. The second inequality follows the fact that $\phi_2(s_2)$ is non-decreasing. The third and forth equalities follow the design of $\phi_1(s_1)$ and $\phi_2(s_2)$ in Theorem 6.

Combining the analysis of the above three cases, Corollary 6 follows.

## G Proof of Proposition 3

We first evaluate the value of $G(v_1^*, v_2^*)$ at its critical points, that is, at the points where $\frac{\partial G(v_1^*, v_2^*)}{\partial v_1^*} = 0$ and $\frac{\partial G(v_1^*, v_2^*)}{\partial v_2^*} = 0$, and show that $G(v_1^*, v_2^*) \geq 0$ holds at these critical points. After that, the

proposition follows by evaluating the values of $G(v_1^*, v_2^*)$ at the four boundary hyperplanes of its domain.

First, let us compute $\frac{\partial G(v_1^*, v_2^*)}{\partial v_1^*}$. It follows that:

$$\frac{\partial G(v_1^*, v_2^*)}{\partial v_1^*}$$

$$= \frac{1}{\alpha_S^*(2)} - \frac{2}{\alpha_S^*(2)} \cdot \frac{v_2^* - c_1}{v_1^* - c_1} + \frac{2}{\alpha_S^*(2)} \cdot \frac{v_2^* - c_2}{v_1^* - c_1} \cdot \phi_2^{-1}(v_2^*).$$

Setting the right-hand side of above equation to be zero, we have

$$\phi_2^{-1}(v_2^*) \cdot (v_2^* - c_2) = v_2^* - c_1 - \frac{v_1^* - c_1}{2}.$$

Using the equation above, we then compute $G(v_1^*, v_2^*)$ at the points that $\frac{\partial G(v_1^*, v_2^*)}{\partial v_1^*} = 0$, it follows that:

$$G(v_1^*, v_2^*)$$

$$= \frac{v_1^* - v_2^*}{\alpha_S^*(2)} + (v_2^* - c_1) \cdot (1 - \phi_1^{-1}(v_1^*))$$

$$+ (v_2^* - c_1 - \frac{v_1^* - c_1}{2}) \cdot \phi_1^{-1}(v_1^*)$$

$$= \frac{v_1^* - v_2^*}{\alpha_S^*(2)} + (v_2^* - c_1) - \frac{v_1^* - c_1}{2} \cdot \phi_1^{-1}(v_1^*)$$

$$\geq \frac{v_1^* - v_2^*}{\alpha_S^*(2)} + (v_2^* - c_1) - \frac{v_1^* - c_1}{2}$$

$$= v_1^* \cdot (\frac{1}{\alpha_S^*(2)} - \frac{1}{2}) + v_2^* \cdot (1 - \frac{1}{\alpha_S^*(2)}) - \frac{c_1}{2}$$

$$\geq \frac{v_1^* - c_1}{2}$$

$$> 0,$$

leading to the conclusion that $G(v_1^*, v_2^*) \geq 0$ holds at its critical points.

Next, we consider the boundary hyperplanes and prove that $G(v_1^*, v_2^*)$ is positive in all four boundary planes given below:

- $G(L_1, v_2^*)$, $\quad \forall v_2^* \in [L_2, U_2]$.
- $G(U_1, v_2^*)$, $\quad \forall v_2^* \in [L_2, U_2]$.
- $G(v_1^*, L_2)$, $\quad \forall v_1^* \in [L_1, U_1]$.
- $G(v_1^*, U_2)$, $\quad \forall v_1^* \in [L_1, U_1]$.

We start with the first one $G(L_1, v_2^*)$:

$$G(L, v_2^*)$$

$$= \frac{2 \cdot L - c_1 - c_2}{\alpha_S^*(2)} + (v_2^* - c_2) - \frac{L + v_2^* - c_1 - c_2}{\alpha_S^*(2)}$$

$$= (v_2^* - c_2) - \frac{v_2^* - L}{\alpha_S^*(2)}$$

$$\geq 0, \qquad \forall L_2 \leq v_2^* \leq U_2,$$

where the equations above follow since $L \geq c_2$ holds (the assumption that the marginal production costs are always less than the valuations).

For the second one $G(U_1, v_2^*)$:

$$G(U_1, v_2^*)$$

$$= \frac{2 \cdot U_1 - c_1 - c_2}{\alpha_S^*(2)} + (v_2^* - c_2) \cdot \phi_2^{-1}(v_2^*) - \frac{U_1 + v_2^* - c_1 - c_2}{\alpha_S^*(2)}$$

$$= (v_2^* - c_2) \cdot \phi_2^{-1}(v_2^*) - \frac{v_2^* - U_1}{\alpha_S^*(2)}$$

$$\geq 0, \qquad \forall L_2 \leq v_2^* \leq U_2 = U.$$

The equations above follow since $(v_2^* - c_2) \cdot \phi_2^{-1}(v_2^*) \geq \int_{s_2=0}^{\phi_2^{-1}(v^*)} (\phi_2(s_2) - c_2) \cdot ds_2 \geq 2 \cdot \frac{v_2^* - U_1}{\alpha_S^*(2)}$ based on the definition of $\phi_2(s)$.

For the third one $G(v_1^*, L_2)$:

$$G(v_1^*, L_2)$$

$$= \frac{2 \cdot v_1^* - c_1 - c_2}{\alpha_S^*(2)} + (L_2 - c_1) \cdot \phi_1^{-1}(v_1^*) - \frac{v_1^* + L_2 - c_1 - c_2}{\alpha_S^*(2)}$$

$$= (L_2 - c_1) \cdot (1 - \phi_1^{-1}(v_1^*)) - \frac{L_2 - v_1^*}{\alpha_S^*(2)}$$

$$\geq 0, \qquad \forall L_1 \leq v_1^* \leq U_1,$$

where the above equation follows since $(L_2 - c_1) \cdot (1 - \phi_1^{-1}(v_1^*)) \geq \int_{s_1=\phi_1^{-1}(v_1^*)}^{\phi_1^{-1}(L_2)} (\phi_1(s_1) - c_1) \cdot ds_1 \geq 2 \cdot \frac{L_2 - v_1^*}{\alpha_S^*(2)}$ (based on the definition of $\phi_1(s)$).

Finally, for the last one $G(v_1^*, U_2)$:

$$G(v_1^*, U_2)$$

$$= \frac{2 \cdot v_1^* - c_1 - c_2}{\alpha_S^*(2)} + (U_2 - c_1) \cdot (1 - \phi_1^{-1}(v_1^*))$$

$$\qquad + (U_2 - c_2) \cdot \phi_1^{-1}(v_1^*) - \frac{v_1^* + U_2 - c_1 - c_2}{\alpha_S^*(2)}$$

$$\geq (U_2 - c_2) - \frac{U_2 - v_1^*}{\alpha_S^*(2)}$$

$$\geq 0, \qquad \forall L = L_1 \leq v_1^* \leq U_1,$$

where the equations above follow since $v_1^* \geq c_2$ holds (again, the assumption that the marginal production costs are always less than the valuations).

Combining all the above analysis, we thus complete the proof of Proposition 3.

# H  Extension of the Lower Bound Results to General Production Cost Functions

In this section, we extend our lower bound result in Theorem 3, originally developed for the high-value case,[4] to general cumulative production cost functions.

Before presenting the main theorem on obtaining a lower bound for general cost functions, let us introduce some notations. Define $f^*(v) : [L, U] \rightarrow \mathbb{R}$ as the conjugate of the total production cost function, where $f^*(v) = \max_{i \in [k]} (v \cdot i - f(i))$. Additionally, let

---

[4]This corresponds to the case when $c_k < L$, or equivalently, the lowest possible valuation $L$ is no less than the highest marginal production cost $c_k$.

$g(v)$ be defined as

$$g(v) = (f^*)'(v) = \sum_{i \in [k]} \mathbb{1}_{\{v \geq c_i\}},$$

where $\mathbb{1}_{\{A\}}$ is the standard indicator function. Let $\underline{k}$ denote the smallest natural number such that:

$$\sum_{i=1}^{k} (L - c_i) > \frac{1}{\alpha} \cdot f^*(L).$$

Following Theorem 3, we also define $\xi$ as follows:

$$\xi = \frac{\frac{1}{\alpha} \cdot f^*(L) - \sum_{i=1}^{\underline{k}-1} (L - c_i)}{L - c_{\underline{k}}}$$

Theorem 7 below extends our lower bound results to settings with general cost functions.

THEOREM 7. Given $S = \{L, U, f\}$ for the OSDoS problem with $k \geq 1$ and general production cost functions $f$, no online algorithm, including those with randomization, can achieve a competitive ratio smaller than $\alpha_S^*(k)$, where $\alpha_S^*(k)$ is the solution to the following system of equations of $\alpha$:

$$\int_{\eta=L}^{u_{\underline{k}}} \frac{g(\eta)}{\alpha \cdot (\eta - c_{\underline{k}})} d\eta = 1 - \xi, \tag{17}$$

$$\int_{\eta=\ell_i}^{u_i} \frac{g(\eta)}{\alpha \cdot (\eta - c_i)} d\eta = 1, u_i = \ell_{i+1}, \ i = \underline{k} + 1, \ldots, k, \tag{18}$$

$$u_k = U. \tag{19}$$

PROOF. The proof proceeds similarly to the proof of Theorem 3 until the derivation of Eq. (5). Given the arrival instance $\mathcal{I}^{(\epsilon)}$ up to the end of stage-$v$, the objective of the offline optimal algorithm equals $f^*(v)$. Therefore, we reformulate Eq. (4) as follows:

$$\text{ALG}\left(\mathcal{I}_v^{(\epsilon)}\right) \geq \frac{1}{\alpha} \cdot f^*(v), \quad \forall v \in [L, U].$$

In the case of general production cost functions, we derive the following inequality to capture the production level changes of an $\alpha$-competitive algorithm:

$$\sum_{i=1}^{k} \psi_i(L) \cdot (L - c_i) +$$

$$\sum_{i=1}^{k} \int_{\eta=L}^{v} (\eta - c_i) d\psi_i(\eta) \geq \frac{1}{\alpha} \cdot f^*(v). \tag{20}$$

In addition, we define $\alpha_S^*(k)$ as follows:

$$\alpha_S^*(k) = \inf \Big\{ \alpha \geq 1 \Big| \text{there exist a set of } k \text{ allocation}$$

$$\text{functions } \{\psi_i(v)\}_{\forall i \in [k]} \in \Omega \text{ that satisfy Eq. (20)} \Big\}.$$

From this point onward, the proof continues in the same manner as the proof of Theorem 3. Let us define the function $\chi^\alpha(v) : [L, U] \rightarrow [0, k]$ and the set of functions $\{\psi_i^\alpha(v)\}_{i \in [k]}$ as specified in Eq. (6) and Eq. (2). Consequently, Lemma 2 holds as long as we have increasing marginal production costs (i.e., diseconomies of scale) and Lemma 3 that follows the definition of $\{\psi_i^\alpha(v)\}_{i \in [k]}$ holds in this case as well.

The primary distinction between the two proofs arises in the following proposition, which gives an explicit design of the function $\{\psi_i^\alpha\}_{\forall i \in [k]}$ by replacing the inequality with an equality in Eq. (20).

PROPOSITION 4. *For any $\alpha \geq \alpha_{\mathcal{S}}^*(k)$, there exist a unique set of functions $\{\psi_i^\alpha(v)\}_{\forall i \in [k]}$ that satisfy Eq. (20) with an equality:*

$$\psi_i^\alpha(v) = 1, \quad \forall v \in [L, U], \ 1 \leq i \leq \underline{k} - 1,$$

$$\psi_{\underline{k}}^\alpha(v) = \begin{cases} 0 & v \leq \ell_{\underline{k}}, \\ \xi + \int_{\eta=L}^v \frac{g(\eta)}{\alpha \cdot (\eta - c_i)} d\eta, & v \in [L, u_{\underline{k}}], \\ 1 & v \geq u_{\underline{k}}, \end{cases}$$

$$\psi_i^\alpha(v) = \begin{cases} 0 & v \leq \ell_i, \\ \int_{\eta=\ell_i}^v \frac{g(\eta)}{\alpha \cdot (\eta - c_i)} d\eta, & v \in [\ell_i, u_i], \\ 1 & v \geq u_i, \end{cases} \quad i = \underline{k} + 1, \ldots, k - 1.$$

$$\psi_{\underline{k}}^\alpha(v) = \begin{cases} 0 & v \leq \ell_{\underline{k}}, \\ \int_{\eta=\ell_{\underline{k}}}^v \frac{g(\eta)}{\alpha \cdot (\eta - c_{\underline{k}})} d\eta, & v \in [\ell_{\underline{k}}, U], \end{cases}$$

*where the intervals are specified by:*

$$\int_{\eta=L}^{u_{\underline{k}}} \frac{g(\eta)}{\alpha \cdot (\eta - c_{\underline{k}})} d\eta = 1 - \xi, \tag{21}$$

$$\int_{\eta=\ell_i}^{u_i} \frac{g(\eta)}{\alpha \cdot (\eta - c_i)} d\eta = 1, u_i = \ell_{i+1}, \ \forall i = \underline{k}, \ldots, k. \tag{22}$$

In the proposition above, for any given $\alpha \geq \alpha_{\mathcal{S}}^*(k)$, the values of $u_i$ and $\ell_i$ can be determined. We begin by solving Eq. (21) to find the value of $u_{\underline{k}}$, and then proceed to find the value of other variables $\{u_i\}_{\forall i}$ using Eq. (22).

Based on the above proposition, as the value of $\alpha$ decreases, the value of $u_{\underline{k}}$ also decreases. Again, following the same reasoning as the proof of Theorem 3, the lower bound $\alpha_{\mathcal{S}}^*(k)$ is the value of $\alpha$ for which $u_{\underline{k}}$ computed above is equal to $U$. We thus complete the proof of Theorem 7. □

## I Extension of the Upper Bound Results to General Production Cost Functions

In this section, we extend the randomized dynamic pricing scheme r-Dynamic, originally developed for the high-value case, to general cumulative production cost functions.

THEOREM 8. *Given $\mathcal{S} = \{L, U, f\}$ for the OSDoS problem with $k \geq 1$, r-Dynamic (Algorithm 1) is $\max_{i \in [k]} \alpha_{\mathcal{S}}^*(k) \cdot (1 + \frac{U_i - c_i}{f^*(U_{i-1})})$-competitive for the following design of the pricing functions $\{\phi_i\}_{\forall i \in [k]}$, where $\alpha_{\mathcal{S}}^*(k)$ is the lower bound obtained in Theorem 7:*

$$\phi_i(s) = L, \quad \forall s \in [0, 1], \ i \in [\underline{k}^* - 1],$$

$$\phi_{\underline{k}^*}(s) = \begin{cases} L & s \in [0, \xi^*], \\ \psi_{\underline{k}}^{-1}(s) & s \in (\xi^*, 1], \end{cases}$$

$$\phi_i(s) = \psi_i^{-1}(s), \quad \forall s \in [0, 1], \ i = \underline{k}^* + 1, \ldots, k,$$

*where the set of functions $\{\psi_i\}_{\forall i \in \{\underline{k}^*, \cdots, k\}}$ are defined as follows:*

$$\psi_{\underline{k}^*}(v) = \xi^* + \int_{\eta=L}^v \frac{g(\eta)}{\alpha \cdot (\eta - c_i)} d\eta, \quad \forall v \in [L, U_{\underline{k}^*}],$$

$$\psi_i(v) = \int_{\eta=\ell_i}^v \frac{g(\eta)}{\alpha \cdot (\eta - c_i)} d\eta, \quad \forall v \in [L_i, U_i], \ i = \underline{k}^* + 1, \ldots, k;$$

*the parameters $\underline{k}^*$ and $\xi^*$ are respectively the values of $\underline{k}$ and $\xi$ defined in Appendix H, corresponding to $\alpha = \alpha_{\mathcal{S}}^*(k)$, and the price intervals $\{[L_i, U_i]\}_{\forall i \in [k]}$ are given as follows:*

$$\int_{\eta=L}^{U_{\underline{k}^*}} \frac{g(\eta)}{\alpha \cdot (\eta - c_{\underline{k}})} d\eta = 1 - \xi,$$

$$\int_{\eta=L_i}^{U_i} \frac{g(\eta)}{\alpha \cdot (\eta - c_i)} d\eta = 1, u_i = \ell_{i+1}, \forall i = \underline{k}^* + 1, \ldots, k.$$

PROOF. The proof will follow the same process as the proof in Appendix E. So we only provide a brief proof sketch.

Consider an arbitrary arrival instance $\mathcal{I} = \{v_t\}_{t \in [T]}$. Recall that the random price vector $\mathbf{P} = \{P_1, \cdots, P_k\}$ is generated using the pricing functions $\{\phi_i\}_{\forall i \in [k]}$ at the beginning of r-Dynamic (line 3 of Algorithm 1). Let us define the random variable $W(\mathbf{P})$, the variable $\omega$ and the price vector $\boldsymbol{\pi}$, the set $\{v_i, \tau_i\}_{\forall i \in [\omega]}$, and $W^{\tau_\omega}(\mathbf{P})$ in the same fashion as in Appendix E.

Following the same reasoning, the property in Eq. (16) can be derived for $\{v_i\}_{\forall i \in [\omega]}$, and the lemmas 7, 8, and 9 follow as well.

We also define $\mathcal{B} \subseteq \mathcal{I}$, as before, to be the set of highest-valued buyers to whom the offline optimal algorithm allocates a unit of the item in instance $\mathcal{I}$. We further divide $\mathcal{B}$ into two subsets: $\mathcal{B}_1$ and $\mathcal{B}_2$, as done in the previous proof. Additionally, we partition $\mathcal{B}_1$ into two subsets: $\mathcal{B}_{1,1}$ and $\mathcal{B}_{1,2}$, as before.

We continue our analysis for two separate cases that can arise depending on the instance $\mathcal{I}$. In this proof, we only provide the proof for the first case and the proof of the second case follows similarly as Appendix E.

**Case 1:** In this case, no buyer in $\mathcal{B}_2$ has a valuation greater than $U_{\omega-1}$ except for the buyer at time $\tau_\omega$. Therefore, the buyer at time $\tau_\omega$ possesses the highest valuation in instance $\mathcal{I}$. The following upper bound can be derived for $\text{OPT}(\mathcal{I})$, which denotes the objective value of the offline optimal algorithm:

$$\text{OPT}(\mathcal{I}) = V(\mathcal{B}_1) + V(\mathcal{B}_2) - \sum_{i=1}^{|\mathcal{B}|} c_i$$

$$\leq V(\mathcal{B}_1) + (|\mathcal{B}_2| - 1) \cdot U_{\omega-1} + v_{\tau_\omega} - \sum_{i=1}^{|\mathcal{B}|} c_i$$

$$= V(\mathcal{B}_{1,1}) + V(\mathcal{B}_{1,2}) + (|\mathcal{B}_2| - 1) \cdot U_{\omega-1} + v_{\tau_\omega} - \sum_{i=1}^{|\mathcal{B}|} c_i$$

$$\leq |\mathcal{B}_{1,1}| \cdot U_{\omega-1} + (V(\mathcal{B}_{1,1}) - |\mathcal{B}_{1,1}| \cdot U_{\omega-1})$$

$$+ |\mathcal{B}_{1,2}| \cdot U_{\omega-1} + (|\mathcal{B}_2| - 1) \cdot U_{\omega-1} + v_{\tau_\omega} - \sum_{i=1}^{|\mathcal{B}|} c_i$$

$$= (|\mathcal{B}_{1,1}| + |\mathcal{B}_{1,2}| + |\mathcal{B}_2| - 1) \cdot U_{\omega-1} + v_{\tau_\omega}$$

$$+ (V(\mathcal{B}_{1,1}) - |\mathcal{B}_{1,1}| \cdot U_{\omega-1}) - \sum_{i=1}^{|\mathcal{B}|} c_i$$

$$\leq f^*(U_{\omega-1}) + (V(\mathcal{B}_{1,1}) - |\mathcal{B}_{1,1}| \cdot U_{\omega-1}) + v_{\tau_\omega} - c_\omega,$$

where the first inequality follows the condition of **Case 1**. The second inequality follows the definition of the sets $\mathcal{B}_{1,1}$ and $\mathcal{B}_{1,2}$. Finally, the third inequality follows since based on definition of $f^*$, we have $(|\mathcal{B}_{1,1}| + |\mathcal{B}_{1,2}| + |\mathcal{B}_2| - 1) \cdot U_{\omega-1} - \sum_{i=1}^{|\mathcal{B}|-1} c_i \leq f^*(U_{\omega-1})$.

Moving forward, we can lower bound the expected performance of r-Dynamic under $\mathcal{I}$, denoted by $\mathbb{E}[\mathrm{ALG}(\mathcal{I})]$, using the same approach as before.

$$\mathbb{E}[\mathrm{ALG}(\mathcal{I})]$$
$$\geq \sum_{i=1}^{\omega-1} \int_0^1 \phi_i(\eta)d\eta + (V(\mathcal{B}_{1,1}) - |\mathcal{B}_{1,1}| \cdot U_{\omega-1}) - \sum_{i=1}^{\omega-1} c_i.$$

Based on the definition of $\{\phi_i\}_{\forall i \in [k]}$, we have:

$$\mathbb{E}[\mathrm{ALG}(\mathcal{I})]$$
$$\geq \sum_{i=1}^{\omega-1} \int_0^1 \phi_i(\eta)d\eta - \sum_{i=1}^{\omega-1} c_i + (V(\mathcal{B}_{1,1}) - |\mathcal{B}_1| \cdot U_{\omega-1})$$
$$= \sum_{i=1}^{k} \psi_i(L) \cdot (L - c_i) + \sum_{i=1}^{\omega-1} \int_{\eta=L}^{U_{\omega-1}} (\eta - c_i)d\psi_i(\eta)$$
$$+ (V(\mathcal{B}_{1,1}) - |\mathcal{B}_1| \cdot U_{\omega-1}).$$

Furthermore, based on the design of $\{\psi_i\}_{\forall i \in [k]}$ in Theorem 8, we have

$$\sum_{i=1}^{k} \psi_i(L) \cdot (L - c_i) + \sum_{i=1}^{\omega-1} \int_{\eta=L}^{U_{\omega-1}} (\eta - c_i)d\psi_i(\eta)$$
$$+ (V(\mathcal{B}_1) - |\mathcal{B}_1| \cdot U_{\omega-1})$$
$$\geq \frac{1}{\alpha_S^*(k)} f^*(U_{\omega-1}) + (V(\mathcal{B}_1) - |\mathcal{B}_1| \cdot U_{\omega-1}).$$

Putting together the above lower and upper bounds, it follows that:

$$\frac{\mathrm{OPT}(\mathcal{I})}{\mathbb{E}[\mathrm{ALG}(\mathcal{I})]}$$
$$\leq \frac{f^*(U_{\omega-1}) + (V(\mathcal{B}_{1,1}) - |\mathcal{B}_{1,1}| \cdot U_{\omega-1}) + v_{\tau_\omega} - c_\omega}{\frac{1}{\alpha_S^*(k)} f^*(U_{\omega-1}) + (V(\mathcal{B}_1) - |\mathcal{B}_1| \cdot U_{\omega-1})}$$
$$\leq \frac{f^*(U_{\omega-1}) + v_{\tau_\omega} - c_\omega}{\frac{1}{\alpha_S^*(k)} f^*(U_{\omega-1})}$$
$$= \alpha_S^*(k) \cdot \left(1 + \frac{v_{\tau_\omega} - c_\omega}{f^*(U_{\omega-1})}\right)$$
$$\leq \alpha_S^*(k) \cdot \left(1 + \frac{U_\omega - c_\omega}{f^*(U_{\omega-1})}\right)$$
$$\leq \max_{i \in [k]} \alpha_S^*(k) \cdot \left(1 + \frac{U_i - c_i}{f^*(U_{i-1})}\right).$$

**Case 2:** In the set of buyers $\mathcal{B}_2$, there are other buyers with valuations greater than $U_{\omega-1}$ besides the buyer at time $\tau_\omega$. The proof in this case follows the same structure as the proof above and the proof in Appendix E. $\square$

