# OpenReview forum: "Posted Price Mechanisms for Online Allocation with Diseconomies of Scale"
_ACM.org/TheWebConf/2025/Conference — WWW 2025 Poster_

### Official Review · Reviewer_6uuC · 2024-11-20

**Novelty:** 4
**Technical Quality:** 4

**Review:**

Pros:
1. The paper addresses an important problem in online resource allocation. The theoretical results provide valuable insights into the design of efficient algorithms for OSDoS.

Cons:
1. There are too many formulas and characters in the article. Readers will easily get lost in them.
2. Some proof seems unnecessary to be included in the main text, such as section 3.3 and section 4.3.

**Questions:**

I noticed that in section 4.2, the authors used a simulation experiment approach. I am curious if there are some real-world examples, such as using actual data from cloud computing systems or other web application to prove that r-Dynamic performs better.

**Reviewer Confidence:**

1: The reviewer's evaluation is an educated guess

**Scope:**

3: The work is somewhat relevant to the Web and to the track, and is of narrow interest to a sub-community

---

### Official Review · Reviewer_HZoP · 2024-11-26

**Novelty:** 5
**Technical Quality:** 4

**Review:**

This paper studies an online selling problem where the cost function increases as the number of sold items increases. They give a bound on the approximation ratio that any mechanism can achieve compared with the optimal offline algorithm. They also design an algorithm that asymptotically achieves this bound and demonstrates it performance against previous works in numerical experiments. The improvement of the bound is due the the new technique it applied. The contribution of this paper is mainly theoretical.

Strength
1. This paper is theoretically non-trivial and novel. They apply a new approach from previous work to improve the bound. Moreover, the same technique helps them to generate an optimal algorithm.
2. The characteristic of the paper is general with an arbitrary increasing cost function.
3. The paper is in general well motivated and presented.

Weakness:
1. Question on soundness. I am confused about your definition of $\mathcal{I}^{(\epsilon)}$ and $\Psi(v)$ in Section 3.2.2. Given that $\Psi(v)$ is defined on a determined instance of $\mathcal{I}^{(\epsilon)}$ (which means constant $\epsilon > 0$ is fixed), it should only be well-defined on $v = L + t * \epsilon$ for integer $t$ but not all $v \in [L, U]$. This becomes a problem when you use the continuity of $\psi(v)$. Also, how does Lemma1 on continuity imply strictly increasing on $v$ in Lines 484-486?
2. The optimal algorithm proposed is not practical. It needs to solve $\alpha^*_S(k)$ and lower $k$ which has not closed form and seems difficult even for pre-calculation.

Minors:
1. The $\epsilon$ in $\mathcal{I}^{(\epsilon)}$ should be restricted into rational numbers (or even $\frac{U-L}{t}$ for integer $t$) to be well-defined.
2. The competitive ratio is not defined when introduced. I was confused in the beginning about whether it was ALG / OPT or OPT / ALG.
3. I feel the paper can do a more direct comparison with the previous work, like ``what previous work has not done and we did it''.

**Questions:**

1. Please address the question on soundness in Weakness 1.
2. Structure of Proof of Theorem 3. To my understanding, Theorem 3 constructs a set of $\psi(v)^{(\alpha)}$ and shows that it is feasible if and only if Equation 1 holds. How does this imply the lower bound? I think the proof mentions some ''uniqueness'', but I don't understand how this uniqueness helps. The unique $\psi(v)^{(\alpha)}$ is constructed from some $\psi(v)$ that also satisfies Equation 5. Why can't $\psi(v)$ imply a better bound?
3. I don't quite get what Lemma 3 does in the proof. It sounds like something seemingly straightforward that the algorithm should allocate the previous item before allocating the next item. This seems trivial as the items are identical except for the cost. I think I misunderstood something here.
4. Is there an efficient way to compute the pricing functions in the algorithm?
5. Is there any high-level idea for Equation 1? I like you explanation for Equation 2 and 3 and hope there is one for Equation 1 as well.

**Reviewer Confidence:**

2: The reviewer is willing to defend the evaluation, but it is likely that the reviewer did not understand parts of the paper

**Scope:**

3: The work is somewhat relevant to the Web and to the track, and is of narrow interest to a sub-community

---

### Official Review · Reviewer_WioX · 2024-11-28

**Novelty:** 5
**Technical Quality:** 6

**Review:**

Summary:
This paper considers an online k-selection problem with increasing marginal production cost. There are two main contributions in this paper: (1) New lower bound is derived for this problem if k is given by designing a family of hard instances. (2) A randomized algorithm is designed to compute an asymptotically tight competitive ratio. In addition, some empirical results are provided to show the performance of the proposed algorithm.

Strength:
(1) The techniques used in lower bound examples and analysis of randomized algorithm are novel, e.g., the design of allocation functions.
(2) The whole paper is well organized and easy to read.


Comments:
In the assumption 1, the buyers’ valuations are bounded in [L, U]. Initially, I think L is used to bound the largest production cost c_k. However, you have shown that in the analysis of lower bound and randomized algorithm, the maximum production cost c_k may not be necessarily lower than L. So, is it possible that, we can derive the lower bound and competitive ratio only with the assumption that we only know the upper range of buyer’s valuation U and the lower range of buyer’s valuation L is unknown?

**Questions:**

Please see comment.

**Reviewer Confidence:**

3: The reviewer is confident but not certain that the evaluation is correct

**Scope:**

3: The work is somewhat relevant to the Web and to the track, and is of narrow interest to a sub-community

---

### Official Review · Reviewer_ygDg · 2024-12-02

**Novelty:** 4
**Technical Quality:** 6

**Review:**

**Summary:**

This paper deals with the online $k$-selection problem with diseconomies of scale (OSDoS), where the goal is to maximize social welfare by controlling the pricing for sequentially arriving buyers. First, they analyze how hard OSDoS is. After that, in order to solve the OSDoS, they propose the d-Dynamic algorithm. They prove that OSDoS is optimal both when $k$ approaches infinity and when $k$ is small but exact $2$. Based on their experiments, r-Dynamic outperforms other algorithms (d-Dynamic and r-Static) in finite $k$.

**Comment:**

First of all, I am not skilled in OSDoS, but I see that they review previous and relevant studies meticulously (the introduction), and their paper is well-organized. I looked over their theorems/lemmas and proofs and recognized that the theoretical contribution is absolutely solid. However, OSDoS has already been studied enormously, and I wonder if this paper just solves the remaining minor problem of some limited settings. Their study is based on an open problem under small inventory $k$. In Fig. 1, however, r-Static, a previous algorithm, looks to outperform r-Dynamic when $k$ is small. Rather, r-Dynamic looks an upper compatible with d-Dynamic, another previous algorithm. (However, their further experiments (Fig. 2) demonstrate that r-Dynamic outperforms r-Static even in small $k$.) In addition, they present that r-Dynamic is optimal in small $k$, but r-Dynamic seems not optimal in Fig. 1, again. Thus, I have many questions as follows.

**Questions:**

**Q)** They claim that the small inventory (small $k$) setting is unresolved (Lines 117-121). Then, what real-world situation does correspond to the case of small $k$ in online allocation?
***
**Q)** Also, if it exists, I recommend they cite previous papers that address the case of small $k$ as an open problem.
***
**Q)** Their experiments show that r-Dynamic has a great advantage in specific cases "when low-valued buyers arrive before high-valued buyers”, but what real-world situation does correspond to the case?
***
**Q)** This paper emphasizes the results that r-Dynamic is theoretically optimal in $k=2$ (Cor. 6) and experimentally near-optimal (Fig. 2). Are these results novel? Otherwise, is r-Static optimal in small $k$? Maybe I missed some description about the optimality of r-Static in the introduction.
***
**Q)** The discrepancy between the results of Fig. 1 and 2 confuses me. Fig. 1 shows that r-Static outperforms r-Dynamic in the small inventory, such as $k=10$. However, Fig. 2 (considering $k=10$) shows that r-Dynamic outperforms r-Static in any buyer-arrival patterns (a)-(c). Why do the results of Fig. 1 and Fig. 2 hold simultaneously? What is the buyer-arrival pattern considered in Fig. 1?
***
**Q)** Several typos:

- “y The above…” in Line 892.

- “d-Dynamic [17]” is maybe [23].

**Reviewer Confidence:**

2: The reviewer is willing to defend the evaluation, but it is likely that the reviewer did not understand parts of the paper

**Scope:**

3: The work is somewhat relevant to the Web and to the track, and is of narrow interest to a sub-community

---

### Official Review · Reviewer_Kd4A · 2024-12-03

**Novelty:** 5
**Technical Quality:** 5

**Review:**

The paper addresses the online allocation problem with diseconomies of scale (OSDoS), where a seller aims to maximize social welfare by pricing items for sequentially arriving buyers, considering increasing marginal production costs. This is a significant problem in various online platforms, including cloud computing and network routing. The authors propose a novel randomized dynamic pricing mechanism, r-Dynamic, which provides a tighter lower bound on the competitive ratio compared to prior work. The primary contribution of this paper is the development of r-Dynamic, a randomized dynamic pricing scheme that outperforms existing online mechanisms in both small and large inventory settings. The authors have successfully bridged the gap between theoretical optimality and practical application by providing a mechanism that is not only theoretically sound but also empirically superior to existing designs. The paper also derives a new tight lower bound for the OSDoS problem, which is a significant theoretical contribution to the field.

Advantages:
(1) The approach this paper proposed is innovative;
(2) The paper is well-organized, with a logical flow that makes it easy to follow. The figures and tables are clear and support the narrative effectively;
(3) The paper proves solid and detailed theoretical contributions;

Disvantages:
(1) The experiments is relatively weak;
(2) While the paper provides a strong theoretical framework, it could benefit from a more in-depth discussion on the practical challenges of implementing the r-Dynamic mechanism in real-world settings;
(3) While the technical depth is a strength, the complexity of the mathematical analysis may be challenging for readers from diverse backgrounds, potentially reducing the accessibility of the paper.

**Questions:**

(1) The data used in the experiments are generated. Comparing to simulated experiments, experiments base on real-world datasets could be more persuasive;
(2) In the experiments, the proposed algorithms should be compared with algorithms from more existing related studies to enhance the persuasiveness of the experimental results;
(3) It is unclear how the proposed mechanism scales with the size of the inventory or the number of buyers. An analysis of scalability could provide additional insights into its practical applicability.

**Reviewer Confidence:**

3: The reviewer is confident but not certain that the evaluation is correct

**Scope:**

4: The work is relevant to the Web and to the track, and is of broad interest to the community